# Randomized Strategic Facility Location with Predictions

**Eric Balkanski**
Columbia University, IEOR
eb3224@columbia.edu

**Vasilis Gkatzelis**
Drexel University, Computer Science
gkatz@drexel.edu

**Golnoosh Shahkarami**
Max Planck Institut für Informatik, Universität des Saarlandes
gshahkar@mpi-inf.mpg.de

## Abstract

In the strategic facility location problem, a set of agents report their locations in a metric space and the goal is to use these reports to open a new facility, minimizing an aggregate distance measure from the agents to the facility. However, agents are strategic and may misreport their locations to influence the facility's placement in their favor. The aim is to design truthful mechanisms, ensuring agents cannot gain by misreporting. This problem was recently revisited through the learning-augmented framework, aiming to move beyond worst-case analysis and design truthful mechanisms that are augmented with (machine-learned) predictions. The focus of this prior work was on mechanisms that are deterministic and augmented with a prediction regarding the optimal facility location. In this paper, we provide a deeper understanding of this problem by exploring the power of randomization as well as the impact of different types of predictions on the performance of truthful learning-augmented mechanisms. We study both the single-dimensional and the Euclidean case and provide upper and lower bounds regarding the achievable approximation of the optimal egalitarian social cost.

## 1   Introduction

In the classic facility location problem the goal is to determine the ideal location for a new facility, taking as input the preferences of a group of $n$ agents. Due to the wide variety of applications that this problem captures, it has received a lot of attention from different perspectives. A notable example is the strategic version of this problem which is motivated by the fact that the participating agents may be able to strategically misreport their preferences, leading to a facility location choice that they prefer over the one that would be chosen if they were truthful. In the strategic facility location problem, the goal is to design *truthful* mechanisms, i.e., mechanisms that elicit the agents' true preferences by carefully removing any incentive for the agents to lie [Moulin, 1980, Procaccia and Tennenholtz, 2013]. This problem has played a central role in the broader literature on mechanism design without money and a lot of prior work has focused on optimizing the quality of the returned location subject to the truthfulness constraint (see Section 1.2 for a brief overview). However, this work has mostly focused on worst-case analysis, often leading to unnecessarily pessimistic impossibility results.

Aiming to overcome the limitations of worst-case analysis, a surge of work during the last few years has focused on the design of algorithms in the *learning-augmented framework* [Mitzenmacher and Vassilvitskii, 2022]. According to this framework, the designer is provided with some machine-learned prediction regarding the instance at hand and the goal is to leverage this additional information in the design process. However, crucially, this information is not guaranteed to be accurate and can,

in fact, be arbitrarily inaccurate. The goal is to achieve stronger performance guarantees whenever the prediction happens to be accurate (known as the *consistency* guarantee), while at the same time maintaining some worst-case guarantee even if the prediction is arbitrarily inaccurate (known as the *robustness* guarantee).

Agrawal et al. [2022] and Xu and Lu [2022] recently introduced the learning-augmented framework to mechanism design problems involving self-interested strategic agents with private information, and both of these works studied the strategic facility location problem. Agrawal et al. [2022] considered both the egalitarian and the utilitarian social cost objectives and provided tight bounds on the best-feasible robustness and consistency trade-off for truthful mechanisms augmented with a prediction regarding what the optimal facility location would be. However, the achievable trade-offs between robustness and consistency may heavily depend on the specific type of prediction that the mechanism is augmented with: note that the consistency guarantee only binds if the prediction is accurate, so more refined predictions bind on fewer instances (the subset of instances where even the refined prediction is accurate) and could, therefore, enable the design of learning-augmented mechanisms with improved guarantees. This leaves open the question of whether mechanisms equipped with more refined predictions can, indeed, achieve better trade-offs between robustness and consistency. Also, Agrawal et al. [2022] restricted their attention to deterministic mechanisms, leaving open the possibility for truthful randomized mechanisms to achieve even stronger guarantees. In this work we significantly expand our understanding of this problem by studying both the power of randomization and the power of alternative predictions.

## 1.1 Our Results

We revisit the problem of designing truthful learning-augmented mechanisms for the strategic facility location problem. These mechanisms ask each agent to report their preferred location in some metric space (which is private information) and they are also augmented with some (unreliable) prediction related to this private information. Using the agents' reported locations, along with the prediction, the mechanism chooses a facility location in this metric space, aiming to (approximately) minimize some social cost measure. Our focus in this paper is on the egalitarian social cost, i.e., the maximum distance between an agent's preferred location and the location chosen for the facility. Our goal is to evaluate the performance of mechanisms that can leverage randomization, as well as the power of different types of predictions.

We first consider the well-studied single-dimensional version of the problem, where all agents lie on a line and the mechanism needs to choose a facility location on that line. Prior work on the strategic facility location problem without predictions, showed that for this class of instances no deterministic truthful mechanism can achieve an approximation better than 2 and no randomized truthful[1] mechanism can achieve an approximation better than 1.5; both of these results are shown to be tight [Procaccia and Tennenholtz, 2013]. In the learning-augmented framework, Agrawal et al. [2022] showed there exists a truthful deterministic mechanism provided with a prediction regarding the optimal facility location that achieves the best of both worlds: a perfect consistency of 1 and an optimal robustness of 2.

Our main result for the single-dimensional version shows that even if the mechanism is provided with the strongest possible prediction (i.e., a prediction regarding every agent's preferred location), any randomized truthful mechanism that is $(1 + \delta)$-consistent for some $\delta \in [0, 0.5]$ can be no better than $(2 - \delta)$-robust. This implies that the previously proposed deterministic learning-augmented mechanism that is 1-consistent and 2-robust and the randomized non-learning-augmented mechanism that is 1.5-robust (and hence also 1.5-consistent) are both Pareto optimal among all randomized mechanisms, even if they are augmented with the strongest possible predictions. Beyond these two extreme points, this result proves a lower bound for the achievable trade-off between robustness and consistency for any randomized mechanism, even with the strongest predictions. We complement this bound by observing that this trade-off can, in fact, be achieved by a truthful randomized mechanism that chooses between the optimal learning-augmented deterministic mechanism and the optimal non-augmented randomized mechanism, thus verifying that our bound fully characterizes this Pareto frontier.

---

[1]A truthful randomized mechanism guarantees truthfulness in expectation. See Section 2 for more details.

We then consider the two-dimensional case, for which much less is known about randomized mechanisms. The best-known truthful randomized mechanism is the Centroid, introduced by Tang et al. [2020], which guarantees a $2-1/n$ approximation. We provide a truthful randomized mechanism that is equipped with a prediction regarding the identities of the most extreme agents (those who would incur the maximum cost in the optimal solution) and, using this prediction, our mechanism achieves 1.67 consistency and 2 robustness. The idea is to use these predictions to select at most three extreme agents, apply the Centroid mechanism to them, and guarantee a 1.67 approximation if the predictions are accurate not only for these three agents but for all other agents as well, utilizing the properties of the Euler line.

An interesting observation is that the lower bound of 1.5 from the single-dimensional case does not extend to two dimensions, so prior work does not imply any lower bound for two dimensions! We prove a lower bound of 1.118 for all truthful randomized mechanisms and then that no deterministic mechanism can simultaneously guarantee better than 2 consistency and better than $1 + \sqrt{2}$ robustness, even if it is augmented with the strongest predictions. Similarly, no truthful randomized mechanism can simultaneously guarantee 1 consistency and better than 2 robustness. The former result proves the optimality of a previously introduced 1-consistent and $1 + \sqrt{2}$-robust truthful deterministic mechanism provided only with a prediction regarding the optimal facility location [Agrawal et al., 2022]. We also show that the latter is tight.

## 1.2 Related Work

**Strategic facility location.**   Moulin [1980] provided a characterization of deterministic truthful facility location mechanisms on the line and introduced the median mechanism, which returns the median of the agent location profile $\mathbf{x} = \langle x_1, \cdots, x_n \rangle$. The median mechanism is known to be truthful, providing an optimal solution for the Utilitarian Social Cost and achieving a 2-approximation for the Egalitarian Social Cost, as demonstrated by Procaccia and Tennenholtz [2013]. Notably, Procaccia and Tennenholtz [2013] showed that this 2-approximation factor represents the best performance achievable by any deterministic truthful mechanism. Building on this, Border and Jordan [1983] extended these results to the Euclidean space by employing median schemes independently in each dimension. Subsequently, Barberà et al. [1993] further generalized this outcome to any $L_1$-norms. Other settings explored include general metric spaces [Alon et al., 2010] and d-dimensional Euclidean spaces [Meir, 2019, Walsh, 2020, El-Mhamdi et al., 2023, Goel and Hann-Caruthers, 2020], as well as circles [Alon et al., 2010, Meir, 2019] and trees [Alon et al., 2010, Feldman and Wilf, 2013]. Fundamental results in truthful facility location often focus on characterizing the space of truthful mechanisms. For the one-dimensional case, Moulin's characterization [Moulin, 1980] reveals that all deterministic truthful mechanisms belong to the "general median mechanisms (GCM)" family. For the two-dimensional case, a similar characterization was provided by Peters et al. [1993]. For a comprehensive review of previous work on this problem, see the survey by Chan et al. [2021].

**Learning-augmented algorithms.**   Worst-case analysis on its own is often not informative enough, and several alternative measures have been proposed to overcome its limitations [Roughgarden, 2021]. Learning-augmented algorithms, or algorithms with predictions [Mitzenmacher and Vassilvitskii, 2022], aim to address these limitations by incorporating predictions into the algorithm's design. Lykouris and Vassilvitskii [2021] studied the online caching problem and introduced two main metrics, consistency and robustness, to evaluate the performance of these algorithms. The online version of the facility location problem, introduced by Meyerson [2001], involves points arriving online, requiring us to assign them irrevocably to either an existing facility or open a new facility. The objective is to minimize the distance of each agent to the assigned facility along with the cost of opening the facilities. Almanza et al. [2021], Jiang et al. [2022], and Fotakis et al. [2021] studied this problem augmented with predictions regarding the location of the optimal facility for each incoming point. We direct interested readers to a curated and frequently updated list of papers in this area [Lindermayr and Megow].

**Learning-augmented mechanism design.**   The framework of learning-augmented mechanism design was first introduced by Agrawal et al. [2022] and Xu and Lu [2022]. Agrawal et al. [2022] focused on the strategic facility location problem, proposing mechanisms that leverage predictions to improve performance while maintaining truthfulness. Their work achieves the best possible consistency and robustness trade-off given a prediction of the optimal facility location. They also

evaluated the performance of their mechanisms as a function of the prediction error. More recently, Christodoulou et al. [2024] provided performance bounds based on an alternative measure of prediction error. Barak et al. [2024] studied this problem assuming that the predictions are regarding each agent's location, but a small fraction of these predictions may be arbitrarily inaccurate. Meanwhile, Chen et al. [2024] evaluated non-truthful mechanisms with respect to their price of anarchy. Istrate and Bonchiş [2022] and Fang et al. [2024] studied the obnoxious facility location version of this problem, aiming to design truthful mechanisms. Apart from the facility location problem, other works in learning-augmented mechanism design have been conducted in various contexts, including strategic scheduling [Xu and Lu, 2022, Balkanski et al., 2023a], auctions [Xu and Lu, 2022, Balkanski et al., 2023b,a, Lu et al., 2023, Caragiannis and Kalantzis, 2024, Gkatzelis et al., 2024], bicriteria mechanism design [Balcan et al., 2024], graph problems with private input [Colini-Baldeschi et al., 2024], and equilibrium analysis [Gkatzelis et al., 2022, Istrate et al., 2024]. Balkanski et al. [2023b] brought together the line of work on learning-augmented mechanism design with the literature on online algorithms with predictions by studying online mechanism design with predictions.

## 2 Preliminaries

In the single facility location problem, there are $n$ strategic agents with a location profile denoted by $\mathbf{x} = \langle x_1, \cdots, x_n \rangle$, where $x_i$ corresponds to the location of agent $i$. A mechanism $f(\mathbf{x})$ outputs a, potentially randomized, location for the facility. The cost incurred by agent $i$ is the expected Euclidean distance $\mathbb{E}[d(x_i, f(\mathbf{x}))]$ between the facility location $f(\mathbf{x})$ and their location $x_i$.

Two renowned cost functions considered in this scenario are Egalitarian Social Cost and Utilitarian Social Cost. In this work, the goal is to optimize the Egalitarian Social Cost $C(f, \mathbf{x}) = \mathbb{E}[\max_{x_i \in \mathbf{x}} d(x_i, f(\mathbf{x}))]$, representing the expected maximum cost experienced by a single agent for each possible outcome of $f(\mathbf{x})$. The optimal location for the facility in the two-dimensional Euclidean space corresponds to the center of the smallest circle that encloses all the points, denoted by $o(\mathbf{x})$.

To minimize the social cost, a mechanism needs to ask the agents to report their preferred locations, $\mathbf{x} \in \mathbb{R}^{2n}$, and then use this information to determine the facility location $f(\mathbf{x}) \in \mathbb{R}^2$. However, the preferred location $x_i$ of each agent $i$ is private information, and they can choose to misreport their preferred location if that can reduce their own cost. A mechanism $f : \mathbb{R}^{2n} \to \mathbb{R}^2$ is considered truthful when no individual agent can benefit by misreporting their location, i.e., for all instances $\mathbf{x} \in \mathbb{R}^{2n}$, every agent $i \in [n]$, and every deviation $x_i' \in \mathbb{R}^2$, we have that $d(x_i, f(\mathbf{x})) \leq d(x_i, f(\mathbf{x}_{-i}, x_i'))$, where $\mathbf{x}_{-i} = \langle x_1, \cdots, x_{i-1}, x_{i+1}, \cdots, x_n \rangle$ is the vector of the locations of all agents except agent $i$.

In the context of randomized mechanisms, we can distinguish between two forms of truthfulness: universally truthful and truthful in expectation. A universally truthful mechanism involves a randomization over deterministic truthful mechanisms, with weights that may depend on the input. On the other hand, a mechanism is considered truthful in expectation if truth-telling yields the agent the maximum expected value. Specifically, a mechanism $f : \mathbb{R}^{2n} \to \mathbb{R}^2$ is truthful in expectation if for all instances $\mathbf{x} \in \mathbb{R}^{2n}$, every agent $i \in [n]$, and every deviation $x_i' \in \mathbb{R}^2$, we have that $\mathbb{E}[d(x_i, f(\mathbf{x}))] \leq \mathbb{E}[d(x_i, f(\mathbf{x}_{-i}, x_i'))]$.

We focus on mechanisms that are both unanimous and anonymous. A mechanism is unanimous if, when all points $\mathbf{x}$ are located at the same position ($x_i = x_j$ for all $i, j \in [n]$), it places the facility at that specific location, i.e., $f(\mathbf{x}) = x_i$. To ensure bounded robustness, a mechanism must be unanimous, as the optimal cost is zero when the facility is at the same location as all points, whereas placing it elsewhere incurs a positive cost. A mechanism is anonymous if its outcome is independent of the agents' identities, meaning it remains unchanged under any permutation of the agents. Finally, we also assume the mechanism is scale-independent, i.e., if we multiply every coordinate of every agent by the same factor, the coordinate of the chosen facility location are scaled in the same way; this captures the fact that it is only the relative distances that really matter in this problem.

Learning-augmented algorithms encompass a class of algorithms that enhance their decision-making process by integrating pre-computed predictions or forecasts. These predictions, obtained from various sources, like statistical models, serve as inputs without requiring real-time learning from new data. For instance, in the single facility location problem, one might consider predictions $\hat{\mathbf{x}}$ regarding all of the agent's preferred locations, a prediction $F^*$ regarding the optimal facility location, or a prediction regarding the identities of the most extreme agents $\hat{\mathbf{e}}$ (the ones that suffer the maximum

cost in the optimal solution), which, as demonstrated in this paper, proves to be quite useful. We denote mechanisms enhanced with predictions in general as $f(\mathbf{x}, *)$. Specifically, for each prediction setting like $\hat{\mathbf{x}}$ or $F^*$, we denote a mechanism enhanced with $\hat{\mathbf{x}}$ and $F^*$ by $f(\mathbf{x}, \hat{\mathbf{x}})$ and $f(\mathbf{x}, F^*)$, respectively. We define mechanisms enhanced with other kinds of predictions in a similar way.

The effectiveness of learning-augmented mechanisms is evaluated using their consistency and robustness. If $\mathbf{x} \lhd *$ denotes all instances $\mathbf{x}$ for which prediction $*$ is accurate, a mechanism is

- $\alpha$-*consistent* if it achieves an $\alpha$-*approximation* when the prediction is correct, i.e.,

$$\max_{\mathbf{x}, *: \mathbf{x} \lhd *} \left\{ \frac{C(f(\mathbf{x}, *), \mathbf{x})}{C(o(\mathbf{x}), \mathbf{x})} \right\} \leq \alpha.$$

- $\beta$-*robust* if it maintains a $\beta$-*approximation* regardless of the quality of the prediction, i.e.,

$$\max_{\mathbf{x}, *} \left\{ \frac{C(f(\mathbf{x}, *), \mathbf{x})}{C(o(\mathbf{x}), \mathbf{x})} \right\} \leq \beta.$$

## 3 Results for the Line

In this section, we provide a lower bound regarding the performance of any randomized mechanism for the line, even if it is equipped with the strongest type of prediction, i.e., a prediction $\hat{\mathbf{x}}$ regarding the preferred location of every agent.

**Theorem 1.** *No mechanism for the line that is truthful in expectation and guarantees $1 + \delta$ consistency for some $\delta \in [0, 0.5]$ can also guarantee robustness better than $2 - \delta$, even if it is provided with full predictions $\hat{\mathbf{x}}$ containing each of the agents' locations.*

The proof consists of two main parts. We first show that, for any instance involving two agents, we can without loss of generality restrict our attention to a class of mechanisms that we call ONLYM mechanisms. We then show the desired lower bound for ONLYM mechanisms.

We introduce the following notations specific to this section. Let $x_L$ be the leftmost reported location and $x_R$ be the rightmost reported location on the line. Therefore, we have $x_L \leq x_R$. Let $M$ denote the midpoint of these two extreme points, i.e., $M = (x_L + x_R)/2$, which would also correspond to the optimal facility location. For simplicity, we sometimes write $f(x_1, \ldots, x_n)$, dropping the angle brackets $\langle \rangle$. ONLYM is the class of mechanisms that, whenever they choose a location within the interval $(x_L, x_R)$, then this location is always $M$.

**Definition 1** (ONLYM mechanisms). *A mechanism $f$ for the line is an* ONLYM *mechanism if $P[f(\mathbf{x}) \in (x_L, x_R) \setminus \{M\}] = 0$.*

The main lemma for the proof of Theorem 1 is the following reduction that allows to restrict our attention to ONLYM mechanisms. The complete proof of Lemma 1 can be found in Appendix A.

**Lemma 1.** *For any problem instance involving two agents with reported locations $\mathbf{x} = \langle x_L, x_R \rangle$ on the line, and any randomized truthful in expectation mechanism achieving $\alpha$-consistency and $\beta$-robustness over this class of instances, there exists a randomized* ONLYM *mechanism that is truthful in expectation and achieves the same consistency and robustness guarantees.*

*Proof Sketch.* Consider a randomized mechanism $f(x_L, x_R)$ with probabilities $p_\ell = \mathbb{P}[f(x_L, x_R) \in (x_L, M)]$ and $p_r = \mathbb{P}[f(x_L, x_R) \in (M, x_R)]$, representing the chances of selecting a location in $(x_L, M)$ and $(M, x_R)$, respectively. If $p_\ell = p_r = 0$, the mechanism already satisfies the desired property. Otherwise, for $p_\ell > 0$ and $p_r > 0$, define the expected locations $\pi_\ell = \mathbb{E}[f(x_L, x_R) \mid f(x_L, x_R) \in (x_L, M)]$ and $\pi_r = \mathbb{E}[f(x_L, x_R) \mid f(x_L, x_R) \in (M, x_R)]$, which can be expressed as convex combinations $\pi_\ell = q_\ell x_L + (1 - q_\ell)M$ and $\pi_r = q_r x_R + (1 - q_r)M$ for some $q_\ell, q_r \in (0, 1)$.

We then construct a new mechanism $f'$ that modifies $f$ by reassigning probability masses to $x_L$, $M$, and $x_R$ as follows:

$$\mathbb{P}[f'(x_L, x_R) = x] = \begin{cases} \mathbb{P}[f(x_L, x_R) = x] & \text{if } x < x_L \text{ or } x > x_R, \\ 0 & \text{if } x \in (x_L, M) \cup (M, x_R), \\ \mathbb{P}[f(x_L, x_R) = x] + q_\ell p_\ell & \text{if } x = x_L, \\ \mathbb{P}[f(x_L, x_R) = x] + (1 - q_\ell)p_\ell + (1 - q_r)p_r & \text{if } x = M, \\ \mathbb{P}[f(x_L, x_R) = x] + q_r p_r & \text{if } x = x_R. \end{cases}$$

To show $f'$ retains the consistency and robustness guarantees of $f$, we show that the expected costs of $f$ and $f'$ are identical for instances with two agents (Lemma 2). Finally, we show that $f'$ maintains the truthfulness in expectation property by verifying that agent costs are unchanged between $f$ and $f'$ (Lemma 3). □

Equipped with Lemma 1, we can now prove Theorem 1.

*Proof of Theorem 1.* We start by proving the result for two-agent instances on the line using any ONLYM mechanism. Then, using Lemma 1, the result follows for any randomized mechanism.

First, note that if the chosen facility location $y$ is at distance $d(M, y)$ from the point $M = (x_L + x_R)/2$, then its egalitarian social cost is equal to $C(o(\mathbf{x}), \mathbf{x}) + d(M, y)$. To verify this fact, assume without loss of generality that this location is on the left of $M$ and note that its distance from the agent located at $x_R$ is $d(x_R, M) + d(M, y)$. As a result, the expected social cost of a mechanism $f$ with agent locations $\mathbf{x}$ is

$$C(f, \mathbf{x}) = C(o(\mathbf{x}), \mathbf{x}) + \mathbb{E}[d(M, f(\mathbf{x}))]. \tag{1}$$

Now, assume that there exists a mechanism that is $(1 + \delta)$-consistent and better than $(2 - \delta)$-robust. For robustness, this would imply that for every instance $\mathbf{x}$, irrespective of the prediction, the mechanism must guarantee that

$$\frac{C(f, \mathbf{x})}{C(o(\mathbf{x}), \mathbf{x})} < 2 - \delta \ \Rightarrow \ \frac{\mathbb{E}[d(M, f(\mathbf{x}))]}{C(o(\mathbf{x}), \mathbf{x})} < 1 - \delta. \tag{2}$$

For this mechanism to be truthful in expectation, it must ensure that no agent has an incentive to misreport their location. Consider the instance $\mathbf{x} = \langle x_L, x_R \rangle$, and note that the agent at $x_L$ has the option to misreport their location as $x'_L = x_L - d(x_L, x_R)$, which would shift the new midpoint $M'$ to $x_L$ in the new instance $\mathbf{x}' = \langle x'_L, x_R \rangle$. This deviation would double the optimal cost, i.e., $C(o(\mathbf{x}'), \mathbf{x}') = 2 \cdot C(o(\mathbf{x}), \mathbf{x})$. Inequality (2), then guarantees that the expected cost for this agent after the deviation would be at most $2(1 - \delta) \cdot C(o(\mathbf{x}), \mathbf{x})$ since we have

$$\frac{\mathbb{E}[d(M', f(\mathbf{x}'))]}{C(o(\mathbf{x}'), \mathbf{x}')} = \frac{\mathbb{E}[d(x_L, f(\mathbf{x}'))]}{2 \cdot C(o(\mathbf{x}), \mathbf{x})} < 1 - \delta. \tag{3}$$

As a result, to ensure that this agent will not misreport, the mechanism needs to ensure that the expected cost of the agent if they report the truth is strictly less than $2(1 - \delta) \cdot C(o(\mathbf{x}), \mathbf{x})$. If we let $P(\leq L) = \mathbb{P}[f(\mathbf{x}) \leq x_L]$ denote the probability that the chosen location is weakly on the left of $x_L$, and $P(M) = \mathbb{P}[f(\mathbf{x}) = M]$ denote the probability that the chosen location is $M$, then the expected cost of the agent located at $x_L$ is at least $C(o(\mathbf{x}), \mathbf{x})P(M) + 2C(o(\mathbf{x}), \mathbf{x})(1 - P(M) - P(\leq L))$ because the mechanism is an ONLYM mechanism. This is even if we assume that the only chosen facility locations weakly on the left of $x_L$ (and weakly on the right of $x_R$, respectively) are exactly on $x_L$ (and exactly on $x_R$, respectively). Therefore, the mechanism needs to always satisfy

$$C(o(\mathbf{x}), \mathbf{x})(P(M) + 2(1 - P(M) - P(\leq L))) < 2(1 - \delta)C(o(\mathbf{x}), \mathbf{x})$$
$$\Rightarrow \ P(M) + 2(1 - P(M) - P(\leq L)) < 2(1 - \delta) \ \Rightarrow \ P(\leq L) > \delta - \frac{P(M)}{2}. \tag{4}$$

If we consider the same instance $\mathbf{x} = \langle x_L, x_R \rangle$, and assume that the mechanism is also provided with accurate predictions $\hat{x}_L = x_L$ and $\hat{x}_R = x_R$ regarding the agent locations, to guarantee $1 + \delta$ consistency, Inequality (1) implies that

$$\frac{C(f(\mathbf{x}, \hat{\mathbf{x}} = \mathbf{x}), \mathbf{x})}{C(o(\mathbf{x}), \mathbf{x})} \leq 1 + \delta \ \Rightarrow \ \frac{\mathbb{E}[d(M, f(\mathbf{x}, \hat{\mathbf{x}}))]}{C(o(\mathbf{x}), \mathbf{x})} \leq \delta. \tag{5}$$

Finally, if we once again consider the same instance with inaccurate predictions $\hat{x}_L = x_L$ and $\hat{x}_R = x_R + d(x_L, x_R)$, we observe that in this case the agent located at $x_R$ would have the option to instead report $x''_R = x_R + d(x_L, x_R) = \hat{x}_R$, and the predictions would then appear to be accurate, forcing the mechanism to satisfy Inequality (5) in order to maintain the required consistency bound. Since the true location $x_R$ would now coincide with the middle point $M''$ of the misreported instance

$\mathbf{x}'' = \langle x_L, x_R'' \rangle$, this would yield the agent located at $x_R$ who misreported an expected cost of $2\delta \cdot C(o(\mathbf{x}), \mathbf{x})$ since we have

$$\frac{\mathbb{E}[d(M'', f(\mathbf{x}'', \hat{\mathbf{x}}))]}{C(o(\mathbf{x}''), \mathbf{x}'')} = \frac{\mathbb{E}[d(x_R, f(\mathbf{x}, \hat{\mathbf{x}}))]}{2 \cdot C(o(\mathbf{x}), \mathbf{x})} \leq \delta. \tag{6}$$

As a result, to ensure that this agent will not misreport, the mechanism needs to ensure that the expected cost of the agent if they report the truth is at most $2\delta \cdot C(o(\mathbf{x}), \mathbf{x})$. If we once again let $P(\leq L)$ denote the probability that the chosen location is weakly on the left of $x_L$, and $P(M)$ denote the probability that the chosen location is $M$, then the expected cost of the agent located at $x_R$ is at least $C(o(\mathbf{x}), \mathbf{x})P(M) + 2C(o(\mathbf{x}), \mathbf{x})P(\leq L)$. This is even if we once again assume that the only chosen facility locations weakly on the left of $x_L$ (and weakly on the right of $x_R$, respectively) are exactly on $x_L$ (and exactly on $x_R$, respectively). Therefore, the mechanism needs to always satisfy

$$P(M) + 2P(\leq L) \leq 2\delta \quad \Rightarrow \quad P(\leq L) \leq \delta - \frac{P(M)}{2}.$$

However, this contradicts Inequality (4), so we conclude that no mechanism can simultaneously guarantee $(1 + \delta)$ consistency and a robustness better than $(2 - \delta)$. $\qquad\square$

### 3.1 The hardness result for the line is tight

We observe that the lower bound regarding the robustness and consistency trade-off shown in Theorem 1 is actually tight. Specifically, it can be achieved by an appropriate randomization between the optimal deterministic learning-augmented mechanism and the optimal non-learning-augmented randomized mechanism.

**Proposition 1.** *For any $\delta \in [0, 0.5]$, there exists a randomized mechanism on the line which, given prediction $F^*$, is truthful in expectation, $(1 + \delta)$-consistent, and $(2 - \delta)$-robust.*

As a result, the bound of Theorem 1 precisely captures the optimal robustness consistency trade-off over all truthful in expectation mechanisms for instances on the line.

### 3.2 Other prediction settings

To gain a more complete picture of different prediction settings, we study an alternative strong prediction that is not strictly stronger than the predicted optimal facility location $F^*$. Specifically, we consider a setting where predictions are available for all pieces of information except for one of the extreme locations. We show that these predictions are not helpful, even without any robustness constraints, to improve the consistency guarantee alone.

**Theorem 2.** *Given a prediction set that provides the identities of all $n$ agents, there exist $n-1$ agents such that, even if we have their exact locations, there is no deterministic truthful mechanism on the line that is better than 2-consistent, and there is no randomized mechanism on the line that is truthful in expectation and better than 1.5-consistent.*

## 4 Results for the Plane

We now consider instances in the Euclidean plane, with a location profile $\mathbf{x} = \langle x_1, \ldots, x_n \rangle$, where $x_i \in \mathbb{R}^2$ for each agent $i$. Missing proofs of this section can be found in Appendix B.

### 4.1 Impossibility Results

Before considering the robustness and consistency guarantees achievable by randomized mechanisms augmented with different types of predictions, we start off by reducing the gap on what is known for mechanisms without predictions.

Since the problem of designing good facility location mechanisms in two dimensions is "harder" than the one-dimensional case, it may seem counterintuitive at first, but the lower bound that prior work proved for all one-dimensional instances does not extend to two dimensions. The reason is that the design space for two-dimensional mechanisms is much richer, and the known lower bounds apply

only to the more restricted class of one-dimensional mechanisms. For example, consider a simple instance with just two agents in two dimensions. For this two-dimensional instance, the mechanism can return a location for the facility that is not on the line containing the two agents' locations, which it, of course, cannot do in one dimension.

Note that, in terms of social cost alone, returning such a location is always Pareto-dominated by returning an appropriate location on the line (e.g., its projection onto the line). However, returning a location that is not on this line also allows the designer to affect the incentives of the participating agents in new and non-trivial ways that are impossible in the one-dimensional case. Specifically, returning points that are not on the line allows us to induce previously impossible cost vectors. For a simple example, in a one-dimensional instance involving two agents at distance 1 from each other, the only facility location that yields the same cost to both agents is the midpoint between them, leading to a cost vector of $(0.5, 0.5)$. However, in two dimensions, we can induce a cost vector of $(c, c)$ for any $c \geq 0.5$ by simply returning the appropriate point on the interval's perpendicular bisector.

This additional flexibility could potentially provide the designer with novel ways to ensure that the agents do not misreport their locations; for example, the classic technique of "money burning," used to achieve incentive compatibility without monetary payments, heavily depends on the designer's ability to penalize and reward agents based on their reports. Although we conjecture that truthful mechanisms are always better off returning a facility on the line containing the agents' locations in this instance, proving such a result appears non-trivial. Due to this additional flexibility in two dimensions, proving inapproximability results for this broader class of mechanisms is a more challenging problem and seems to require new techniques and insights.

**Theorem 3.** *Any randomized mechanism that is truthful in expectation in the Euclidean metric space has an approximation ratio of at least* $1.118$.

Our following two results provide impossibility results for both deterministic and randomized mechanisms augmented with perfect predictions. For deterministic mechanisms, Agrawal et al. [2022] showed that there is a mechanism that, given a prediction about the optimal facility location, is 1-consistent and $(1 + \sqrt{2})$-robust. Our impossibility result for deterministic mechanisms shows that stronger predictions do not help: even with predictions about the location of each agent, there is no deterministic mechanism that achieves a robustness better than $1 + \sqrt{2}$ and a consistency that improves over the best approximation achievable without predictions.

**Theorem 4.** *For any $\delta > 0$, there is no deterministic truthful mechanism that is $(2 - \delta)$-consistent and $\left(1 + \sqrt{2} - \delta\right)$-robust, even if it is provided with full predictions $\hat{\mathbf{x}}$ containing each of the agents' locations.*

The next result shows that there is no hope of achieving the best of both worlds in the randomized setting; to obtain 1 consistency, we would need to sacrifice robustness.

**Theorem 5.** *For any $\delta > 0$, there is no randomized mechanism that is truthful in expectation, 1-consistent, and $(2 - \delta)$-robust, even for two-agent instances and even if it is provided with full predictions $\hat{\mathbf{x}}$ (the location of each agent). This is tight, i.e., there exists a randomized mechanism that is truthful in expectation, 1-consistent, and 2-robust for two-agent instances.*

## 4.2 Positive Results Using Extreme ID Prediction

We now turn to positive results, and provide a learning-augmented randomized mechanism that is provided with a new type of prediction: the prediction does not provide us with any actual location, but instead only provides us with the identities $\hat{\mathbf{e}} = \langle e_1, \cdots, e_k \rangle$ of the $k$ agents who would suffer the maximum cost in the optimal solution (we refer to them as "extreme agents"), i.e., $\{e_1, \ldots, e_k\} = \arg\max_{e_i \in [n]} d(x_{e_i}, o(\mathbf{x}))$. Note that the smallest circle that encloses all the points is the *circumcircle* of the locations of the extreme agents. Therefore, the center of this circle is $o(\mathbf{x})$, the optimum solution.

We propose a mechanism leveraging predictions derived from IDs of extreme agents. The main idea is to return the centroid of the extreme agents, denoted by $\mathcal{G}$ with a probability of half, and each of the extreme points with a probability of $1/2k$ to prevent misreporting incentives.

Tang et al. [2020] run this mechanism over all agents (not only the extreme ones) and achieve a $2 - 1/n$ approximation. We improve the approximation factor (in case of having good predictions) to

---
**Mechanism 1:** Centroid Mechanism on Extreme Agents

---
**Input** :Location profile $\mathbf{x} = \langle x_1, \cdots, x_n \rangle$, Predictions $\hat{\mathbf{e}} = \langle e_1, \cdots, e_k \rangle$
**Output**:Probability distribution $\mathcal{P}$ on location of the facility

With probability $1/2$:
return the centroid $\mathcal{G} = \frac{x_{e_1} + \cdots + x_{e_k}}{k}$
With probability $1/2k$:
return each point $x_{e_1}, \cdots, x_{e_k}$

---

$2 - \frac{1}{3} \approx 1.67$ by running this mechanism only on extreme agents. We use their ideas to maintain truthfulness and we use new techniques to show the approximation guarantees for an arbitrarily number of agents. Moreover, as long as any returned location falls within the minimum enclosing circle of the agents predicted to be extreme (which is the case for Mechanism 1), this ensures an approximation factor of 2 in case of having bad predictions.

First, we argue that $k \geq 2$, meaning that there are at least two extreme agents on the minimum enclosing circle. Otherwise, one can find a smaller circle containing all the points. As a warm-up, we first consider the $k = 2$ case and propose a randomized mechanism that is truthful in expectation and achieves 1.5 consistency and 2 robustness for any number of agents.

**Theorem 6.** *Assume there are only two extreme agents, i.e., $k = 2$. Then, given predictions $\hat{\mathbf{e}} = \langle e_1, e_2 \rangle$ that provides the IDs of the only two extreme agents, there exists a randomized mechanism that is truthful in expectation and achieves 1.5 consistency and 2 robustness for any number of agents.*

We then show that it is sufficient to only consider the case where we have exactly three extreme agents on the minimum enclosing circle, meaning $k = 3$. If we have more than three agents located on the minimum enclosing circle, a continuous perturbation of the points makes the probability of having at least four points lying on a circle infinitesimally small [Berg et al., 2008]. For $k = 3$, we use the properties of the Euler line to prove the 1.67 consistency.

**Theorem 7.** *Given a prediction set that provides the IDs of the extreme agents, there exists a randomized mechanism that is truthful in expectation and achieves 1.67 consistency and 2 robustness for any number of agents.*

*Proof.* Let us first consider instances with only three extreme agents. Mechanism 1 is truthful since other agents apart from $x_{e_1}, x_{e_2}, x_{e_k}$ cannot influence the result and Mechanism 1 is equivalent to the mechanism of Tang et al. [2020] over agents $e_1, \ldots, e_k$ and since this mechanism is truthful in expectation, agents $e_1, \ldots, e_k$ cannot benefit from misreporting their locations. Since the mechanism returns the reported locations, rather than their predictions, and the centroid is guaranteed to be inside the circumcircle, we can conclude that the mechanism has a robustness factor of 2. The technical aspect of this proof involves establishing the consistency guarantee by considering the Euler line.

**Euler line.** Given three arbitrary points $x_1, \ldots, x_3$, their circumcircle is the smallest circle that encloses the three points, their centroid is $(x_1 + x_2 + x_3)/3$, and their orthocenter is the point where the three altitudes (the perpendicular line segments from a vertex to the line that contains the opposite side) intersect. In any triangle, the center of the circumcircle ($O$), the centroid ($\mathcal{G}$), and the orthocenter ($H$) are collinear, forming the Euler line. One property of the Euler line is that $\mathcal{G}$ is positioned midway between $O$ and $H$. Additionally, $d(O, \mathcal{G}) = d(\mathcal{G}, H)/2$, implying $d(O, \mathcal{G}) = d(O, H)/3$.

Let $R$ denote the radius of the circumcircle of $x_{e_1}, x_{e_2}, x_{e_3}$ and assume without loss of generality that the optimum cost is $R = 1$. For any $j \in [n]$, we have

$$d(x_j, \mathcal{G}) \leq d(x_j, O) + d(O, \mathcal{G}) \leq R + d(O, \mathcal{G})$$

where the first inequality is by triangle inequality and the second is because all the points are within the circumcircle of the locations of the extreme agents when the predictions are correct. Since $H$ lies within the circumcircle, we infer $d(O, H) \leq R$, conclusively showing that $d(O, \mathcal{G}) \leq R/3$. Since $R$ is the optimum cost and any agent can reach $O$ by paying that cost, the cost of returning $\mathcal{G}$ for any agent is upper bounded by $R + d(O, \mathcal{G})$, which is at most $1.34R$. Consequently, the approximation of Mechanism 1 in case of having accurate predictions is less or equal than

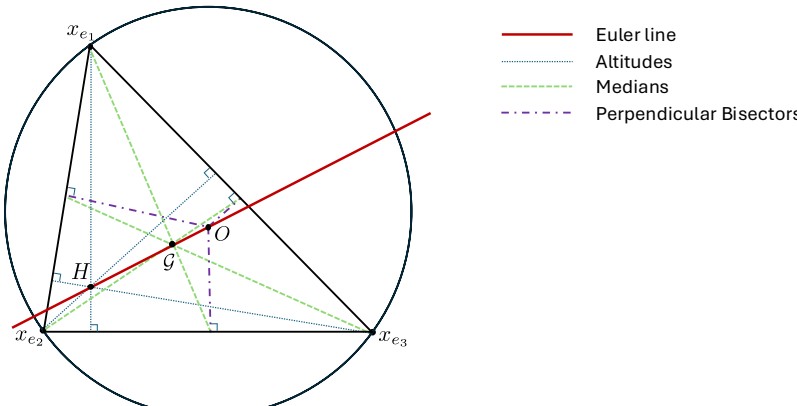

Figure 1: $\mathcal{G} = (x_{e_1} + x_{e_2} + x_{e_3})/3$ is the centroid of $x_{e_1}, \ldots, x_{e_3}$, which is the intersection of the three medians. $H$ is the orthocenter, which is the intersection of the three altitudes, and $O$ is the center of the circumcircle, which is the intersection of the three perpendicular bisectors. In any triangle, the circumcenter ($O$), the centroid ($\mathcal{G}$), and the orthocenter ($H$) are collinear, forming the Euler line. Moreover, $d(O, \mathcal{G}) = \frac{1}{2} d(\mathcal{G}, H)$.

$$\frac{1}{2k} \frac{1}{R} \sum_{i=1}^{k} \max_{j \in [n]} d(x_{e_i}, x_j) + \frac{1}{2} \frac{1}{R} \max_{j \in [n]} d(x_j, \mathcal{G}) \leq 1 + \frac{1}{2} \frac{R + d(O, \mathcal{G})}{R} \leq 1 + \frac{1}{2} \left(1 + \frac{1}{3}\right) \approx 1.67.$$

where the first inequality is since $d(x_{e_i}, x_j) \leq d(x_{e_i}, O) + d(O, x_j) \leq 2R$ when the predictions are correct. Next, we show how we can modify the mechanism to achieve the same approximation guarantees as having three extreme agents while maintaining truthfulness.

As mentioned previously, the key concept involves perturbing the instance before requesting predictions to have at most three extreme agents. Given any instance $\mathbf{x} = \langle x_1, \cdots, x_n \rangle$, define a perturbed instance $\tilde{\mathbf{x}} = \langle \tilde{x}_1, \cdots, \tilde{x}_n \rangle$, which is the result of a continuous perturbation of $\mathbf{x}$. Consider the set of predictions for the IDs of extreme agents in $\tilde{\mathbf{x}}$. Although the extreme agents of the perturbed instance $\tilde{\mathbf{x}}$ might differ from those of the original instance $\mathbf{x}$, their costs remain very close to the maximum cost in case of having good predictions. Therefore, the circumcircle of them is a good representation of the minimum enclosing circle and results in the same approximation guarantees.

Since we run the mechanism on the main instance $\mathbf{x}$, truthfulness holds as before. Note that since we ask for the ID of extreme agents of the perturbed instance $\tilde{\mathbf{x}}$, we have consistency guarantees if predictions are accurate based on the perturbed instance $\tilde{\mathbf{x}}$, and in any case we can ensure the robustness factor of 2. $\qquad \square$

## 5   Future Directions

The problem of designing randomized facility location mechanisms in the Euclidean space is very natural and several open questions remain, even without predictions. While numerous studies have addressed this problem in restricted spaces, there remains a gap regarding between the best possible approximation guarantee in $(1.118, 2 - 1/n)$. The possibility of a truthful in expectation mechanism achieving better than a $2 - 1/n$ approximation is intriguing. Obtaining a stronger lower bound than $1.118$ would also be very interesting.

Augmenting these mechanisms with predictions introduces a new dimension to the problem. While we have explored various prediction settings and provided both positive and negative results, many of the current findings in Euclidean space lack tightness. Finding tight results for different types of predictions would enable meaningful comparisons and help identify which prediction strategy is most effective in different scenarios.

## Acknowledgments

Eric Balkanski and Vasilis Gkatzelis were partially supported by NSF grant CCF-2210502. Eric Balkanski was also supported by NSF grant IIS-2147361. Vasilis Gkatzelis was also supported by NSF CAREER award CCF-2047907.

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

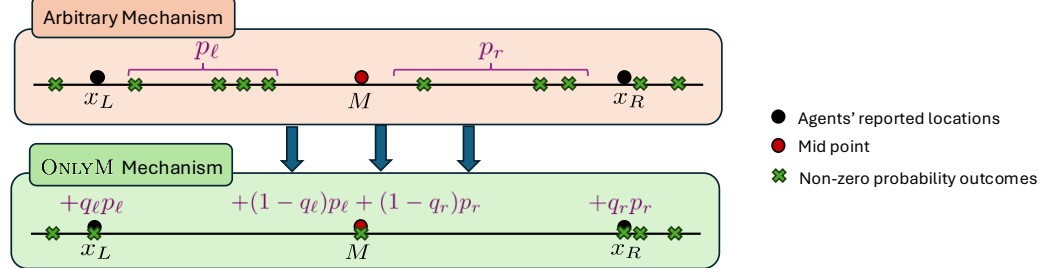

Figure 2: The original mechanism $f(.)$ and the new ONLYM mechanism $f'(.)$ in the proof of Lemma 1, where $p_\ell = \mathbb{P}[f(x_L, x_R) \in (x_L, M)]$ and $p_r = \mathbb{P}[f(x_L, x_R) \in (M, x_R)]$, $\pi_\ell = \mathbb{E}[f(x_L, x_R) \mid f(x_L, x_R) \in (x_L, M)]$ and $\pi_r = \mathbb{E}[f(x_L, x_R) \mid f(x_L, x_R) \in (M, x_R)]$, and $q_\ell$ and $q_r$ are such that $\pi_\ell = q_\ell x_L + (1 - q_\ell)M$ and $\pi_r = q_r x_R + (1 - q_r)M$.

## A  Missing Proof of Section 3

**Lemma 1.** *For any problem instance involving two agents with reported locations $\mathbf{x} = \langle x_L, x_R \rangle$ on the line, and any randomized truthful in expectation mechanism achieving $\alpha$-consistency and $\beta$-robustness over this class of instances, there exists a randomized ONLYM mechanism that is truthful in expectation and achieves the same consistency and robustness guarantees.*

*Proof.* Consider any randomized mechanism $f(x_L, x_R)$ and let $p_\ell = \mathbb{P}[f(x_L, x_R) \in (x_L, M)]$ and $p_r = \mathbb{P}[f(x_L, x_R) \in (M, x_R)]$ represent the probabilities that this mechanism chooses a facility location in $(x_L, M)$ and $(M, x_R)$, respectively. If $p_\ell = p_r = 0$, the mechanism already satisfies the desired property, and the proof is complete. Otherwise, if $p_\ell > 0$, define $\pi_\ell = \mathbb{E}[f(x_L, x_R) \mid f(x_L, x_R) \in (x_L, M)]$ and, if $p_r > 0$, define $\pi_r = \mathbb{E}[f(x_L, x_R) \mid f(x_L, x_R) \in (M, x_R)]$ as the expected locations returned by the mechanism when restricted to $(x_L, M)$ and $(M, x_R)$, respectively.

Since $\pi_\ell$ lies in $(x_L, M)$ and $\pi_r$ lies in $(M, x_R)$, we can express these two points as convex combinations of $x_L$, $M$, and $x_R$: $\pi_\ell = q_\ell x_L + (1 - q_\ell)M$ and $\pi_r = q_r x_R + (1 - q_r)M$ for some $q_\ell, q_r \in (0, 1)$. We then modify the original mechanism $f$ to a new ONLYM mechanism $f'$ defined as

$$\mathbb{P}[f'(x_L, x_R) = x] = \begin{cases} \mathbb{P}[f(x_L, x_R) = x] & \text{if } x < x_L \text{ or } x > x_R \\ 0 & \text{if } x \in (x_L, M) \cup (M, x_R) \\ \mathbb{P}[f(x_L, x_R) = x] + q_\ell p_\ell & \text{if } x = x_L \\ \mathbb{P}[f(x_L, x_R) = x] + (1 - q_\ell)p_\ell + (1 - q_r)p_r & \text{if } x = M \\ \mathbb{P}[f(x_L, x_R) = x] + q_r p_r & \text{if } x = x_R \end{cases}$$

The construction of $f'$ is illustrated in Figure 2. To show that mechanism $f'$ achieves the same consistency and robustness guarantees as mechanism $f$, we show that their expected costs are equal on all instances with two agents.

**Lemma 2.** *For all mechanisms $f$ for the line and instances $\mathbf{x} = \langle x_L, x_R \rangle$ with two agents, the expected costs of $f$ and $f'$ over $\mathbf{x}$ are equal, i.e., $C(f, \mathbf{x}) = C(f', \mathbf{x})$.*

*Proof.* First, note that

$$\mathbb{E}\left[\max_{x_i \in \{x_L, x_R\}} d(x_i, f(\mathbf{x})) \mid f(\mathbf{x}) \in (x_L, M)\right] \cdot \mathbb{P}[f(\mathbf{x}) \in (x_L, M)]$$
$$= \mathbb{E}[d(x_R, f(\mathbf{x})) \mid f(\mathbf{x}) \in (x_L, M)] \cdot p_\ell$$
$$= d(x_R, \pi_\ell) \cdot p_\ell.$$

Similarly, we have

$$\mathbb{E}\left[\max_{x_i \in \{\mathbf{x}\}} d(x_i, f(\mathbf{x})) \mid f(\mathbf{x}) \in (M, x_R)\right] \cdot \mathbb{P}[f(\mathbf{x}) \in (M, x_R)] = d(\pi_r, x_L) \cdot p_r,$$

which implies that

$$C(f', \mathbf{x}) - C(f, \mathbf{x}) = \mathbb{E}\left[\max_{x_i \in \{x_L, x_R\}} d(x_i, f'(\mathbf{x}))\right] - \mathbb{E}\left[\max_{x_i \in \{x_L, x_R\}} d(x_i, f(\mathbf{x}))\right]$$

$$= d(x_R, x_L) \cdot (\mathbb{P}[f'(\mathbf{x}) \in \{x_L, x_R\}] - \mathbb{P}[f(\mathbf{x}) \in \{x_L, x_R\}])$$
$$+ d(x_R, M) \cdot (\mathbb{P}[f'(\mathbf{x}) = M] - \mathbb{P}[f(\mathbf{x}) = M])$$
$$- d(x_R, \pi_\ell) \cdot p_\ell - d(x_L, \pi_r) \cdot p_r$$
$$= d(x_R, x_L) \cdot (q_\ell p_\ell + q_r p_r) + d(x_R, M) \cdot ((1 - q_\ell)p_\ell + (1 - q_r)p_r)$$
$$- d(x_R, \pi_\ell) \cdot p_\ell - d(x_L, \pi_r) \cdot p_r$$
$$= d(x_R, x_L) \cdot (q_\ell p_\ell + q_r p_r) + d(x_R, M) \cdot ((1 - q_\ell)p_\ell + (1 - q_r)p_r)$$
$$- d(x_R, x_L) \cdot q_\ell p_\ell - d(x_R, M) \cdot (1 - q_\ell)p_\ell$$
$$- d(x_L, x_R) \cdot q_r p_r - d(x_L, M) \cdot (1 - q_r)p_r$$
$$= 0.$$

$\square$

Next, to show that mechanism $f'$ is also truthful in expectation, we first show that the costs of the agents do not change between $f$ and $f'$.

**Lemma 3.** *For all mechanisms $f$ for the line and instances $\mathbf{x} = \langle x_L, x_R \rangle$ with two agents, the cost of the agent at location $x_L$ is identical under both $f$ and $f'$, i.e., $\mathbb{E}\left[d(x_L, f(\mathbf{x}))\right] = \mathbb{E}\left[d(x_L, f'(\mathbf{x}))\right]$. Similarly, we have $\mathbb{E}\left[d(x_R, f(\mathbf{x}))\right] = \mathbb{E}\left[d(x_R, f'(\mathbf{x}))\right].$*

*Proof.* By definition of $f'$, we have

$$\mathbb{E}\left[d(x_L, f(\mathbf{x}))\right] - \mathbb{E}\left[d(x_L, f'(\mathbf{x}))\right]$$
$$= d\left(x_L, \mathbb{E}\left[f(\mathbf{x}) \mid f(\mathbf{x}) \in (x_L, M)\right] = \pi_\ell\right) \cdot \mathbb{P}\left[f(\mathbf{x}) \in (x_L, M)\right]$$
$$+ d\left(x_L, \mathbb{E}\left[f(\mathbf{x}) \mid f(\mathbf{x}) \in (M, x_R)\right] = \pi_r\right) \cdot \mathbb{P}\left[f(\mathbf{x}) \in (M, x_R)\right]$$
$$+ d\left(x_L, M\right) \cdot (\mathbb{P}\left[f(\mathbf{x}) = M\right] - \mathbb{P}\left[f'(\mathbf{x}) = M\right])$$
$$+ d\left(x_L, x_R\right) \cdot (\mathbb{P}\left[f(\mathbf{x}) = x_R\right] - \mathbb{P}\left[f'(\mathbf{x}) = x_R\right])$$
$$= d(x_L, q_\ell x_L + (1 - q_\ell)M) \cdot p_\ell + d(x_L, q_r x_R + (1 - q_r)M) \cdot p_r$$
$$- d(x_L, M)((1 - q_\ell)p_\ell + (1 - q_r)p_r) - d(x_L, x_R)(q_r p_r)$$
$$= d(x_L, M)(1 - q_\ell)p_\ell + d(x_L, x_R)q_r p_r + d(x_L, M)(1 - q_r)p_r$$
$$- d(x_L, M)(1 - q_\ell)p_\ell - d(x_L, M)(1 - q_r)p_r - d(x_L, x_R)q_r p_r$$
$$= 0.$$

$\square$

Next, we use the previous lemma to show that mechanism $f'$ preserves the truthful in expectation guarantee of $f$.

**Lemma 4.** *If a mechanism $f$ for the line is truthful in expectation over instances with two agents, then $f'$ is also truthful in expectation over instances with two agents.*

*Proof.* Assume that $f$ is truthful in expectation over instances with two agents. Assume that one of the agents deviates and reports a false location in mechanism $f'$. Due to symmetry, and without loss of generality, assume that the agent located at $x_L$ deviates and reports a false location $x'_L$. We need to show

$$\mathbb{E}\left[d(x_L, f'(x'_L, x_R))\right] \geq \mathbb{E}\left[d(x_L, f'(x_L, x_R))\right].$$

Since the original mechanism $f(x_L, x_R)$ is truthful in expectation, we have

$$\mathbb{E}\left[d(x_L, f(x'_L, x_R))\right] \geq \mathbb{E}\left[d(x_L, f(x_L, x_R))\right]. \tag{7}$$

Combining these two inequalities with Lemma 3, it suffices to show

$$\mathbb{E}\left[d(x_L, f'(x'_L, x_R))\right] \geq \mathbb{E}\left[d(x_L, f(x'_L, x_R))\right].$$

To prove the above inequality, we consider two main cases. First, we focus on the scenario where the agent located at $x_L$ deviates to the right, i.e., $x_L < x'_L$. Let $M' = \frac{x'_L + x_R}{2}$. Assume $x'_L \le x_R$, define $p'_\ell = \mathbb{P}[f(x'_L, x_R) \in (x'_L, M')]$ and $p'_r = \mathbb{P}[f(x'_L, x_R) \in (M', x_R)]$. If $p'_\ell = p'_r = 0$, then $f'(x'_L, x_R) = f(x'_L, x_R)$, and hence $\mathbb{E}[d(x_L, f'(x'_L, x_R))] = \mathbb{E}[d(x_L, f(x'_L, x_R))]$. Therefore, the proof is complete in this case.

Next, consider the distribution where the mechanism $f(x'_L, x_R)$ returns a random facility location within the intervals $(x'_L, M')$ if $p'_\ell > 0$, or within $(M', x_R)$ if $p'_r > 0$. The expected locations within these intervals are $\pi'_\ell \in (x'_L, M')$ and $\pi'_r \in (M', x_R)$, which can be expressed as $\pi'_\ell = q'_\ell x'_L + (1 - q'_\ell)M'$ and $\pi'_r = q'_r x_R + (1 - q'_r)M'$, where $q'_\ell$ and $q'_r$ are the respective convex coefficients.

In the first case, where $x_L < x'_L$, we show that the cost to the left agent is the same across the two mechanisms. The key idea is that the difference between the two mechanisms, in terms of their returned locations, lies within the intervals $(x'_L, M')$ and $(M', x_R)$, both on the right side of $x_L$. Thus, the agent's cost is fully determined by the expected locations in these intervals. Specifically we have

$$
\begin{aligned}
&\mathbb{E}\left[d(x_L, f(x'_L, x_R))\right] - \mathbb{E}\left[d(x_L, f'(x'_L, x_R))\right] \\
&= d\left(x_L, \mathbb{E}\left[f(x'_L, x_R) \mid f(x'_L, x_R) \in (x'_L, M')\right] = \pi'_\ell\right) \cdot \mathbb{P}\left[f(x'_L, x_R) \in (x'_L, M')\right] \\
&\quad + d\left(x_L, \mathbb{E}\left[f(x'_L, x_R) \mid f(x'_L, x_R) \in (M', x_R)\right] = \pi'_r\right) \cdot \mathbb{P}\left[f(x'_L, x_R) \in (M', x_R)\right] \\
&\quad + d\left(x_L, x'_L\right) \cdot \left(\mathbb{P}\left[f(x'_L, x_R) = x'_L\right] - \mathbb{P}\left[f'(x'_L, x_R) = x'_L\right]\right) \\
&\quad + d\left(x_L, M'\right) \cdot \left(\mathbb{P}\left[f(x'_L, x_R) = M'\right] - \mathbb{P}\left[f'(x'_L, x_R) = M'\right]\right) \\
&\quad + d\left(x_L, x_R\right) \cdot \left(\mathbb{P}\left[f(x'_L, x_R) = x_R\right] - \mathbb{P}\left[f'(x'_L, x_R) = x_R\right]\right) \\
&= d(x_L, q'_\ell x'_L + (1 - q'_\ell)M') \cdot p'_\ell + d(x_L, q'_r x_R + (1 - q'_r)M') \cdot p'_r \\
&\quad - d(x_L, x'_L)q'_\ell p'_\ell - d(x_L, M')((1 - q'_\ell)p'_\ell + (1 - q'_r)p'_r) - d(x_L, x_R)q'_r p'_r \\
&= d(x_L, x'_L)q'_\ell p'_\ell + d(x_L, M')(1 - q'_\ell)p'_\ell + d(x_L, x_R)q'_r p'_r + d(x_L, M')(1 - q'_r)p'_r \\
&\quad - d(x_L, x'_L)q'_\ell p'_\ell - d(x_L, M')((1 - q'_\ell)p'_\ell + (1 - q'_r)p'_r) - d(x_L, x_R)q'_r p'_r \\
&= 0.
\end{aligned}
$$

Note that in the case of $x_R < x'_L$, the same arguments hold with a minor change in notation. Define $p'_\ell = \mathbb{P}[f(x'_L, x_R) \in (M', x'_L)]$ and $p'_r = \mathbb{P}[f(x'_L, x_R) \in (x_R, M')]$. Consider the expected locations $\pi'_\ell$ and $\pi'_r$ within the intervals $(M', x'_L)$ and $(x_R, M')$, respectively.

Focusing our attention on the more interesting case where $x'_L < x_L$, we aim to show that $\mathbb{E}[d(x_L, f'(x'_L, x_R))] \ge \mathbb{E}[d(x_L, f(x'_L, x_R))]$. If we focus on the interval $(x'_L, M')$ and assume $x_L \le M'$, the expected location of the facility returned by the two mechanisms, $f(x'_L, x_R)$ and $f'(x'_L, x_R)$, conditioned on the facility being in the interval $(x'_L, M')$, is the same; this is true by construction (our reduction maintains the expected location $\pi'_\ell$ between the reported location and the midpoint). The crucial difference between the two mechanisms is that although $f(x'_L, x_R)$ may return any facility location in the $(x'_L, M')$ interval, the mechanism $f'(x'_L, x_R)$ returns the same expected location using only $x'_L$ and $M'$ in its support. We show that the cost to an agent located at any point $x_L$ in the $(x'_L, M')$ interval is weakly higher in $f'(x'_L, x_R)$ than it is in $f(x'_L, x_R)$, thus maintaining the truthfulness guarantee. Intuitively, this is true because the two mechanisms return the same expected location $\pi'_\ell$ in that interval, but $f'(x'_L, x_R)$ only returns the extreme points of the interval, which hurt the agent at $x_L$ the most.

More formally, we replace any probability assigned to a point in $(x'_L, M')$ with a convex combination between $x'_L$ and $M'$, and each time we do this, we weakly increase the cost to an agent who is located at any point in $(x'_L, M')$. Specifically, for any outcome $y \in (x'_L, M')$ of the mechanism $f(x'_L, x_R)$ and any $x_L \in (x'_L, M')$, if we have $y = w \cdot x'_L + (1 - w) \cdot M'$ then $d(x_L, y) < w \cdot d(x_L, x'_L) + (1 - w) \cdot d(x_L, M')$.

Since $\pi'_\ell = q'_\ell x'_L + (1 - q'_\ell)M'$, at the end of this process, we will end up with $q'_\ell p'_\ell$ probability increase on $x'_L$, and $(1 - q'_\ell)p'_\ell$ probability increase on $M'$. Therefore, we have

$$\mathbb{E}\left[d(x_L, f(x'_L, x_R))\right] < d(x_L, x'_L)q'_\ell p'_\ell + d(x_L, M')\left((1 - q'_\ell)p'_\ell + (1 - q'_r)p'_r\right) + d(x_L, x_R)q'_r p'_r$$
$$= \mathbb{E}\left[d(x_L, f'(x'_L, x_R))\right].$$

In the case where $x_L > M'$, similar arguments apply in the interval $(M', x_R)$. $\qquad\square$

The proof of Lemma 1 then immediately follows from Lemma 2 and Lemma 4.

$\square$

**Proposition 1.** *For any $\delta \in [0, 0.5]$, there exists a randomized mechanism on the line which, given prediction $F^*$, is truthful in expectation, $(1 + \delta)$-consistent, and $(2 - \delta)$-robust.*

*Proof.* We show that the bound of the robustness consistency trade-off in Theorem 1 is tight. To verify the tightness for $\delta = 0.5$, note that it is possible to achieve a robustness of 1.5 (and therefore also a consistency of 1.5) using the following randomized truthful in expectation mechanism of Procaccia and Tennenholtz [2013]:

**Mechanism 1** (LRM). *Given an input $\mathbf{x}$, the mechanism returns $x_L$ with probability $1/4$, $x_R$ with probability $1/4$, and $M$ with probability $1/2$.*

On the other extreme, to verify the tightness of the theorem for $\delta = 0$, note that it is possible to achieve a robustness of 2 with a consistency of 1 using the following truthful deterministic learning-augmented mechanism of Agrawal et al. [2022]:

**Mechanism 2** (MinMaxP). *Given an input $\mathbf{x}$ and a prediction $F^*$ regarding the optimal facility location, the mechanism returns $F^*$ if $F^* \in [x_L, x_R]$, $x_L$ if $F^* < x_L$, and $x_R$ if $F^* > x_R$.*

In fact, for any $\delta \in [0, 0.5]$, we can achieve a robustness of $2 - \delta$ and a consistency of $1 + \delta$ by randomly choosing which one of these two mechanisms to run. Specifically, we can achieve these bounds by running the LRM mechanism with probability $2\delta$ and the MinMaxP mechanism with probability $1 - 2\delta$. Note that since the decision regarding which one of the two truthful mechanisms to run is independent of the agents' reports, the resulting randomized mechanism is truthful as well. $\square$

**Theorem 2.** *Given a prediction set that provides the identities of all $n$ agents, there exist $n - 1$ agents such that, even if we have their exact locations, there is no deterministic truthful mechanism on the line that is better than 2-consistent, and there is no randomized mechanism on the line that is truthful in expectation and better than 1.5-consistent.*

*Proof.* We first address the deterministic case with two agents whose locations are given by $\mathbf{x} = \langle x_L, x_R \rangle$. Assume we have a prediction $\hat{x}_L$, which predicts the location of the leftmost agent. We will show that even if $\hat{x}_L$ is accurate, there is no truthful mechanism with a consistency better than 2.

We use the same argument as in Theorem 3.2 of Procaccia and Tennenholtz [2013]. Assume, for contradiction, that $f : \mathbb{R}^2 \to \mathbb{R}$ is a truthful mechanism with a consistency less than 2. Consider the location profile $\mathbf{x} = \langle x_L = 0, x_R = 1 \rangle$ and the prediction $\hat{x}_L = 0$. To achieve consistency better than 2, the mechanism needs to choose a facility location $y = f(\mathbf{x})$ such that $y \in (0, 1)$.

Now consider an alternative location profile $\mathbf{x}' = \langle x'_L = 0, x'_R = y \rangle$ with the prediction $\hat{x}_L = 0$. The optimal facility location for this profile is at $y/2$, yielding a maximum cost of $y/2$. To maintain consistency better than 2, the mechanism should place the facility within the interval $(0, y)$. However, if this were to occur, the rightmost agent would have an incentive to report a location of 1 instead of $y$, as this lie would place the facility exactly at $y$, rather than within $(0, y)$. This contradicts the truthfulness of the mechanism.

This argument extends to cases with arbitrary $n$ by situating all other agents at 0 in both profiles $\mathbf{x}$ and $\mathbf{x}'$, with accurate predictions for them at 0. Similar reasoning holds true.

For the randomized setting, we use the idea from the proof of Theorem 3.4 in Procaccia and Tennenholtz [2013]. We first focus on the case with two agents and then extend the result to any

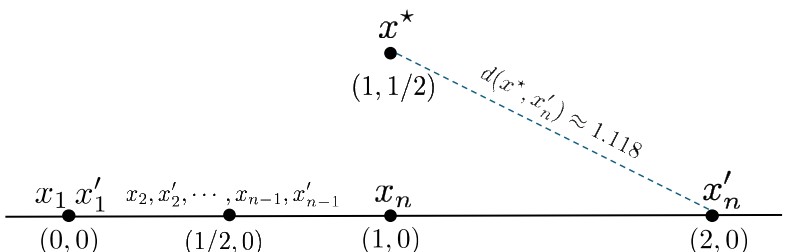

Figure 3: Instances $\mathbf{x} = \langle x_1 = (0,0), x_2 = \cdots = x_{n-1} = (1/2,0), x_n = (1,0)\rangle$, and $\mathbf{x}' = \langle x'_1 = (0,0), x'_2 = \cdots = x'_{n-1} = (1/2,0), x'_n = (2,0)\rangle$ in the proof of Theorem 2. It is assumed that $d(x_n, f(\mathbf{x})) \geq 1/2$ and $x^\star \in \arg\min_{y:d(x_n,y)\geq 1/2} C(y, \mathbf{x})$

.

number of agents. Let $f$ be a randomized truthful in expectation mechanism. Consider the location profile $\mathbf{x} = \langle x_L = 0, x_R = 1\rangle$. We have $f(\mathbf{x}) = \mathcal{P}$, where $\mathcal{P}$ is a probability distribution over $\mathbb{R}$. There exists $x_i \in \mathbf{x}$ such that $\mathbb{E}[d(x_i, \mathcal{P})] \geq 1/2$. If this agent's location prediction is unavailable, we can prove that consistency better than 1.5 cannot be achieved.

Assume $\mathbb{E}[d(x_R, \mathcal{P})] \geq 1/2$ and that we have an accurate prediction $\hat{x}_L = 0$ for the leftmost agent $x_L$. Consider the profile $\mathbf{x}' = \langle x'_L = 0, x'_R = 2\rangle$, with an accurate prediction $\hat{x}_L = 0$. For truthfulness, the expected distance from the location 1 should still be $1/2$, otherwise the rightmost agent would lie in profile $\mathbf{x}$. We know that if $\mathbb{E}[d(M, \mathcal{P})] = \Delta$, then the expected maximum cost is $\Delta + 1$. With $\Delta \geq 1/2$, the expected cost is at least 1.5, whereas the optimal cost is 1, resulting in a consistency of at least 1.5.

This argument extends to arbitrary $n$ by situating all other agents at $1/2$ in both profiles $\mathbf{x}$ and $\mathbf{x}'$, with accurate predictions for them at $1/2$. Similar reasoning holds true. $\qquad\square$

## B  Missing Proofs of Section 4

**Theorem 3.** *Any randomized mechanism that is truthful in expectation in the Euclidean metric space has an approximation ratio of at least* 1.118.

*Proof.* Let $f(\mathbf{x}) = \mathcal{P}$ be a randomized truthful in expectation mechanism, where $\mathcal{P}$ is a probability distribution over $\mathbb{R}^2$. Consider the location profile $\mathbf{x} = \langle x_1 = (0,0), x_2 = \cdots = x_{n-1} = (1/2,0), x_n = (1,0)\rangle$. There exists an $x_i$ (either $x_1$ or $x_n$) such that $d(x_i, f(\mathbf{x})) = \mathbb{E}_{y\sim\mathcal{P}}d(x_i, y) \geq 1/2$. Without loss of generality, assume that $d(x_n, f(\mathbf{x})) \geq 1/2$. Now consider the location profile $\mathbf{x}' = \langle x'_1 = (0,0), x'_2 = \cdots = x'_{n-1} = (1/2,0), x'_n = (2,0)\rangle$. To maintain truthfulness, we must have $d(x_n = (1,0), f(\mathbf{x}')) \geq 1/2$, preventing the agent at location $x_n = (1,0)$ in the profile $\mathbf{x}$ from having an incentive to lie about being at location $x'_n = (2,0)$.

Extending the result of Theorem 3.4 in Procaccia and Tennenholtz [2013] from the line to the Euclidean metric, if we have $d(o(\mathbf{x}), f(\mathbf{x})) \geq \Delta$, then the expected maximum cost is at least

$$\sqrt{\Delta^2 + C(o(\mathbf{x}), \mathbf{x})^2}.$$

Therefore, the expected cost of $\mathbf{x}'$ is at least $\sqrt{\frac{1}{4} + \left(\frac{2}{2}\right)^2}$, resulting in a $\sqrt{1.25} \approx 1.118$ approximation, as the optimum cost is $\frac{d(x'_1, x'_n)}{2} = 1$. $\qquad\square$

**Theorem 4.** *For any $\delta > 0$, there is no deterministic truthful mechanism that is $(2 - \delta)$-consistent and $(1 + \sqrt{2} - \delta)$-robust, even if it is provided with full predictions $\hat{\mathbf{x}}$ containing each of the agents' locations.*

*Proof.* Using the characterization provided by Peters et al. [1993], we know that any deterministic, truthful, anonymous, and unanimous mechanism can be expressed as a Generalized Coordinatewise

Median (GCM) mechanism with $n-1$ constant points in $P$. The GCM mechanism takes as input the reported locations $\mathbf{x}$ of the $n$ agents and a multiset $P$ of fixed points (known as phantom points), which are constant and independent of the agents' reported locations. It then outputs the coordinatewise median of the combined set $\mathbf{x} \cup P$.

Consider a five-agent instance with $\mathbf{x}$ containing four agents at $(0, 20)$ and one agent at $(0, 10)$. Our proof first shows that to achieve a consistency better than 2, this mechanism needs to introduce four constant points all with $y$-coordinates greater than 10. Otherwise, even if the predictions are correct, the $y$-coordinate of $\text{GCM}(\mathbf{x}, P)$ will be at most 10, leading to consistency greater than or equal to 2, since the optimal point would be $(0, 15)$.

Then, we show that if all these four constant points have $y$-coordinates greater than 10, the robustness of the GCM mechanism is at least $1 + \sqrt{2}$. Let $\tilde{x}$ be the median of the $x$-coordinates of the four constant points, and now consider the following instance, where the optimal cost is 1.

Consider a five-agent instance with $\mathbf{x}$ containing three agents at $(\tilde{x}, 0)$, one agent at $(\tilde{x} - 1, 1)$, and one agent at $\left( \tilde{x} - 1 - \frac{1}{\sqrt{2}}, -\frac{1}{\sqrt{2}} \right)$.

Since $\tilde{x}$ is the median of the $x$-coordinates of both true agent locations $\mathbf{x}$ and constant points $P$, $\tilde{x}$ will be the $x$-coordinate of the $\text{GCM}(\mathbf{x}, P)$. Since all the constant points have $y$-coordinates greater than 10, the $y$-coordinate of the $\text{GCM}(\mathbf{x}, P)$ will be 1.

Since $(\tilde{x}, 1)$ has a distance of $1 + \sqrt{2}$ from $\left( \tilde{x} - 1 - \frac{1}{\sqrt{2}}, -\frac{1}{\sqrt{2}} \right)$ while $(\tilde{x} - 1, 0)$, the optimal point, has a distance of 1 from all the points, we can conclude that the robustness factor is at least $1 + \sqrt{2}$.  □

**Theorem 5.** *For any $\delta > 0$, there is no randomized mechanism that is truthful in expectation, 1-consistent, and $(2 - \delta)$-robust, even for two-agent instances and even if it is provided with full predictions $\hat{\mathbf{x}}$ (the location of each agent). This is tight, i.e., there exists a randomized mechanism that is truthful in expectation, 1-consistent, and 2-robust for two-agent instances.*

*Proof.* Consider the instance $\mathbf{x} = \langle x_L = 0, x_R = 2 \rangle$, and assume we have predictions $\hat{\mathbf{x}} = \langle \hat{x}_L = 0, \hat{x}_R = 2 \rangle$. To achieve 1-consistency, a mechanism should return $M = 1$ with probability 1.

Now consider another instance $\mathbf{x}' = \langle x'_L = 0, x'_R = 1 \rangle$ with incorrect predictions $\hat{\mathbf{x}} = \langle \hat{x}_L = 0, \hat{x}_R = 2 \rangle$. To ensure that the rightmost agent will not misreport to $x_R = 2$ and have a cost of zero, the mechanism needs to ensure that the expected cost of the agent if they report the truth is at most zero, which means returning $x'_R$ with probability 1. This results in a robustness factor of 2.

Agrawal et al. [2022] proposed a deterministic Minimum Bounding Box mechanism that is 1-consistent and $1 + \sqrt{2}$-robust. We argue that this mechanism is 1-consistent and 2-robust for two-agent instances.

**Mechanism 3** (Minimum Bounding Box). *Given $n$ points $\mathbf{x} = \langle (x_1, y_1), \cdots, (x_n, y_n) \rangle$, and a prediction $F^* = (x_F, y_F)$, return $(MinMaxP(\langle x_1, \cdots, x_n \rangle, x_F), MinMaxP(\langle y_1, \cdots, y_n \rangle, y_F))$.*

**Observation 1.** *Mechanism 3 is 1-consistent and 2-robust for instances with only two agents.*

*Proof.* One of the two diagonals of the minimum bounding box has two agents on its vertices. Therefore, any point inside or on this box has the approximation factor of 2.  □

Therefore, the upper and lower bounds match for instances with two agents, but it remains unclear where the exact value is even for three agents.

□

**Theorem 6.** *Assume there are only two extreme agents, i.e., $k = 2$. Then, given predictions $\hat{\mathbf{e}} = \langle e_1, e_2 \rangle$ that provides the IDs of the only two extreme agents, there exists a randomized mechanism that is truthful in expectation and achieves 1.5 consistency and 2 robustness for any number of agents.*

*Proof.* If we only have two agents $x_{e_1}$ and $x_{e_2}$ located on the minimum enclosing circle, $x_{e_1}x_{e_2}$ represent a diameter of this circle; otherwise, we can find a smaller circle containing all the points. Therefore, the middle point of these two locations, $(x_{e_1} + x_{e_2})/2$, is the optimum location of the facility. Mechanism 1 runs LRM mechanism on the diameter of the minimum enclosing circle, meaning that it returns the optimum location with a probability of $1/2$ and each of the reported locations of the extreme agents with a probability of $1/4$, resulting in a $1.5$ consistency.

In terms of truthfulness, since no other agent besides $x_{e_1}$ and $x_{e_2}$ impacts the mechanism, it suffices to show that they do not have an incentive to lie. Agents $x_{e_1}$ and $x_{e_2}$ lack such incentive for reasons similar to those in LRM Mechanism. Additionally, as any returned location falls within the minimum enclosing circle, it ensures a robustness factor of 2. $\square$

