# OpenReview forum: "Randomized Strategic Facility Location with Predictions"
_NeurIPS.cc/2024/Conference — NeurIPS 2024 poster_

### Official Review · Reviewer_hDFx · 2024-07-12

**Soundness:** 3
**Presentation:** 3
**Contribution:** 3
**Rating:** 4
**Confidence:** 4

**Summary:**

The paper considers the strategic (single) facility location problem where the goal is to decide on the location of a single facility given the preferred locations {p_i} of n agents. The preferred locations p_i are private information and the goal is to design strategyproof mechanisms so that the agents reveal their true preferences. The quality of a solution is measured by its egalitarian cost, i.e., the maximum distance between any agent location (p_i) and the facility. The paper considers the problem when the mechanism has as input some predictions regarding agent's private preferences. This problem has been initially considered by Agrawal et al in EC'22 where they assume that the prediction is a single point denoting the location of the optimal facility.

The paper mainly shows lower bounds on the achievable consistency / robustness in different settings. First, on a line metric, the authors show that any mechanism that is (1+delta) consistent (for 0 < delta < 0.5) must be >=(2-delta)-robust, even with strong predictions about the preferences of each agent. This trade-off is also tight and every point on this frontier can be attained by a simple mechanism that randomizes between the (1-consistent, 2-robust) mechanism of Agrawal et al and a classical 1.5-robust (and 1.5 consistent) randomized mechanism. Second, in two dimensional Euclidean metrics, the authors first provide a new lower bound of 1.11 in the classical (no predictions) setting. In the learning augmented setting, they show two other lower bounds - (i) no deterministic mechanism can have better than 2 consistency and better than 1+\sqrt(2)-robustness; (ii) no randomized mechanism can have 1-consistency and better than 2-robustness. Lastly, the authors consider a new setup where the mechanism has access to the identities of the most extreme agents, i.e, agents who incur the maximum cost in the optimal solution. In this setting, the paper shows that a variant of the centroid mechanism is 1.67-consistent and 2-robust.

**Strengths:**

- The paper considers a very natural problem. The proposed mechanism is very accessible and easy to follow.
- The paper has been well-written and is nice to read.

**Weaknesses:**

- The paper mainly gives lower bounds in the different settings. The only positive result is obtained via running the centroid mechanism on the (predicted) extreme agents.
- The proofs are terse and a bit hard to follow - for example, adding a small picture would help improve readability of many proofs (e.g. proof of Thm 2 could use a diagram illustrating the pythagoras ineq used).

**Questions:**

- Could you add a discussion / reference for why the 1-d lower bound does not hold in 2 dimensions?

---

> ### Author Rebuttal · Authors · 2024-08-07
>
> Thank you for your helpful comments and suggestions.
>
> * **Reviewer:** "The paper mainly gives lower bounds in the different settings. The only positive result is obtained via running the centroid mechanism on the (predicted) extreme agents."
>
> **Response:** Although many of our results take the form of impossibility results, we do prove that all these lower bounds are tight, i.e., achievable by truthful mechanisms. These tight bounds provide a complete picture for a natural problem, they fully capture the power and limitations of learning-augmented mechanisms in this setting, and they can be very useful information for designers, even beyond their theoretical value.
>
>
> * **Reviewer:** "adding a small picture would help improve readability of many proofs"
>
> **Response:** We agree with the reviewer and we have now enhanced our paper with additional figures: we have added one figure in Section 3 to clarify the proof of Lemma 1, and two figures in Section 4 to make the proofs of Theorem 2 and Theorem 6 easier to follow. You can find these three figures in the PDF file that we uploaded in the general response.
>
>
>  * **Reviewer:** Could you add a discussion / reference for why the 1-d lower bound does not hold in 2 dimensions?
>
> **Response:** This was, indeed, quite surprising for us as well when we first realized it, and it appears like no prior work has made this interesting observation. To provide more intuition regarding this fact, we have now added the following paragraph before Theorem 2. Thank you for the suggestion.
>
>
> > Since the problem of designing good facility location mechanisms in 2-dimensions is ``harder'' than the 1-dimensional one, it may seem counter-intuitive at first, but the lower bound that prior work proved for all 1-dimensional instances does not extend to 2 dimensions. The reason is that the design space for 2-dimensional mechanisms is much richer, and the known lower bounds apply only to the more restricted class of 1-dimensional mechanisms. For example, consider a simple instance with just two agents in two dimensions. For this 2D instance, the mechanism can return a location for the facility that is not on the line that contains the two agent's location, which it, of course, cannot do in one dimension. Note that, in terms of social cost alone, returning such a location is always Pareto dominated by returning an appropriate location on the line (e.g., its projection on the line), but returning a location that is not on this line also allows the designer to potentially affect the incentives of the participating agents in new and non-trivial ways that are impossible in the 1-dimensional case. Specifically, returning points that are not on the line allows us to induce previously impossible cost vectors: for a toy example, note that in a 1-dimensional instance involving two agents at distance 1 from each other, the only facility location that causes the same cost to both agents is the mid-point between them, leading to a cost vector of (0.5, 0.5). However, in 2-dimensions we can induce a cost vector of $(c, c)$ for any $c>0.5$ by just returning the appropriate point on the interval's perpendicular bisector. This additional flexibility could potentially provide the designer with novel ways to ensure that the agents will not lie; e.g., the classic technique of "money burning", used to achieve incentive compatibility without monetary payments, heavily depends on the designer's ability to penalize and reward agents, depending on their reports. Although we conjecture that truthful mechanisms are always better off returning a facility on the line containing the agent's locations in the aforementioned instance, proving such a result appears to be non-trivial and, due to this additional flexibility for the mechanism in two dimensions, proving inapproximability results for this broader class of mechanisms is a harder problem and seems to require new techniques and insights.

---

> ### Author Response · Authors · 2024-08-12
>
> Dear reviewer hDFx,
>
> Thank you once again for your helpful comments and suggestions. As the discussion period is coming to an end soon, it appears like you still maintain a negative score with high confidence, so we were wondering whether our responses did not address all of your concerns, and whether you would like us to provide any additional support for our submission while we still can.

---

### Official Review · Reviewer_MNoo · 2024-07-12

**Soundness:** 3
**Presentation:** 3
**Contribution:** 3
**Rating:** 6
**Confidence:** 3

**Summary:**

In this paper, the authors study the metric space facility location problem with machine-learned predictions. Specifically, they focus on the setting that the mechanism has predictions of every agent’s preferred location and with randomized mechanisms. The goal is to find truthful mechanisms that minimize the egalitarian social cost. The effectiveness of the algorithms are evaluated by their consistency (approximation ratio when the prediction is correct) and robustness (approximation ratio regardless of the quality of the prediction).

For the single-dimensional setting, they first show that prior work with weaker predictions only on the optimal facility location are already Pareto optimal. Then, they further complete the full Pareto frontier by randomizing between the deterministic and randomized mechanisms.

For the two-dimensional setting, they prove an improved lower bound of 1.11 approximate ratio for all randomized mechanisms. They also show that no deterministic mechanism can be better than 2-consistency and (1+$\sqrt{2}$)-robustness, and no randomized mechanism can be better than 1-consistency and (1+$\sqrt{2}$)-robustness. Finally, they show that a truthful randomized mechanism that only has a prediction of the identities of the most suffered agent can achieve 1.5-consistency and 2-robustness.

**Strengths:**

This paper is nicely built on prior work of both the classic facility location problem and mechanism augmentation with machine-learned predictions, which gained much attention in recent years. The paper is mostly well written and I enjoyed reading it. The theoretical results are solid and I think would be valuable for the research community.

**Weaknesses:**

* Several related papers [Procaccia and Tennenholtz, 2013],  [Agrawal et al. 2022], [Tang et al., 2020] study both the total social cost and egalitarian social cost, but this paper only focuses on the egalitarian cost. I understand that the egalitarian cost matches well with the “extreme ID prediction” assumption in Section 4.2, but I think this paper would be much stronger with some results of the total social cost setting.
* Minor comments
  * I was initially confused about line 83-85, as it is first claimed no deterministic mechanism can combine better than 2-consistency with 1 + $\sqrt{2}$-robustness, and then mentioned previous work achieved 1-consistency with 1 + $\sqrt{2}$-robustness. Maybe replace “combine better” with a more concrete statement, like the statement in Theorem 3.
  * Page 2, line 70: 2 + $\delta$ robust -> 2 - $\delta$ robust
  * Page 3, line 121: need a space in “design,Agrawal”
  * Page 5, line 230: “cause at least as much cost to the agent on $x_R$ as to the other agent” Would you explicitly point out what “the other agent” means?

**Questions:**

* In Lemma 1, line 235 assumes the agent at $x_L$ reports a false location. Is this assumption general enough to show that no agent would misreport? Also in Lemma 1, line 238 to 248 considered two cases that $x_L < x’_L$ or  $x_L > x’_L$. How about the case $x’_L > x_R$?
* As mentioned above, several related papers [Procaccia and Tennenholtz, 2013],  [Agrawal et al. 2022], [Tang et al., 2020] study both the total social cost and egalitarian social cost. I wonder if you have explored the total social cost objective.
* Have you considered this problem in a general metric space? Are there any straightforward results?

**Limitations:**

Future directions including closing the gap between upper and lower bounds are discussed in the paper. I think there are also other interesting directions like extending the setting to general metric or consider total social cost and the objective function.

---

> ### Author Rebuttal · Authors · 2024-08-07
>
> Thank you for your helpful comments and suggestions.
>
> * **Reviewer:** Minor comments.
>
> **Response:** We agree with all the suggested changes and have applied all of them.
> Regarding the first comment, we have edited the statement as follows:
>
> > We then turn to mechanisms augmented with the strongest predictions and show that no such deterministic mechanism can simultaneously guarantee better than $2$-consistency and better than $1 + \sqrt{2}$-robust, and no randomized one can simultaneously guarantee $1$-consistency and better than $2$-robustness.
>
>
> * **Reviewer:** "In Lemma 1, line 235 assumes the agent at $x_L$ reports a false location. Is this assumption general enough to show that no agent would misreport? Also in Lemma 1, line 238 to 248 considered two cases that $x_L < x'_L$ or $x_L > x'_L$. How about the case $x_L'> x_R$?"
>
> **Response:** Yes, the analysis of Lemma 1 does imply that neither one of the two agents would misreport. Note that while we focus on the potential misreporting of $x_L$, we could directly apply symmetric arguments for the potential misreporting of the other agent. Furthermore, while we analyze the incentives of the agent located at $x_L$, we do not need to make any assumption regarding the report of the other agent, i.e., we do not need to assume that the other agent reports truthfully. Therefore, the analysis proves that an agent has no incentive to misreport, irrespective of what the other agent reports. Finally, the argument that we provide for the case that $x'_L > x_L$ (i.e., that "the cost of the agent is the same across the two mechanisms'') also applies if $x_L'> x_R$. Specifically, the only thing that this argument uses is that the points that are affected by the lie are all on on the same side of $x_L$ (in this case all the points that are affected have coordinates greater than $x_L$). As a result, the cost of the agent located at $x_L$ is uniquely determined by the expected coordinate value of these points (the ones that lie on its right), and the two mechanisms yield the same expected coordinate value for these points.
>
> Although our analysis is correct and the mechanism is well-defined, we agree with the reviewer that the writing is somewhat confusing (e.g., not being clear enough regarding the distinction between reported locations and true locations), so we plan to revisit the part of the proof that discusses truthfulness and add more details. In the meantime, we would be more than happy to provide the reviewer with any additional details to convince them that there is no issue with the proof (e.g., we would be happy to provide a completely rewritten version of the proof).
>
> * **Reviewer:** "I wonder if you have explored the total social cost objective."
>
> **Response:** We believe that exploring this problem with total social cost is a very interesting direction.
>
> We know that the median mechanism gives the optimum solution in one dimension. Therefore, there is no need for prediction in the line metric. The coordinate-wise median achieves a $\sqrt{2}$ approximation in the Euclidean plane metric, and Agrawal et al. (2022) studied this problem with predictions in the deterministic setting.
>
> Since there are no known randomized mechanisms that strictly improve the total social cost over the best-known deterministic mechanisms, we decided to focus solely on egalitarian social cost (for which there are more studies beyond the line metric, such as Alon et al. (2010), who studied truthful mechanisms on circles and trees with egalitarian social cost) and explore randomized mechanisms with several prediction settings.
>
> Moreover, we believe that for any learning-augmented problem, studying different prediction settings (ideally all possible settings) and establishing different bounds (ideally tight) is important and currently missing from the literature.
>
> * **Reviewer:** "Have you considered this problem in a general metric space?"
>
> **Response:** We believe that exploring this problem in more general metrics is also a very interesting direction. However, we are still a few steps away from addressing it.
>
> Currently, there are no tight results in the randomized setting, even in the Euclidean plane. When no tight bounds are available, it is unclear whether an improvement with predictions is due to the power of predictions alone or if it is also possible without predictions.
>
> The main challenge is the lack of characterization for randomized mechanisms, both in the Euclidean plane and in more general metric spaces. Although we could not find a characterization for randomized mechanisms in the Euclidean plane or make our results (with or without predictions) tight, we have made progress toward this goal.
>
> Moreover, we believe that some of our techniques can be extended to more general metrics. For instance, with more complicated arguments, we could modify our Centroid Mechanism on Extreme Agents for Euclidean space to achieve better than 2-consistency and 2-robustness.

---

> > ### Comment · Reviewer_MNoo · 2024-08-12
> >
> > I thank the authors for their response. I will keep my score and don't have more questions.

---

### Official Review · Reviewer_vBqc · 2024-07-13

**Soundness:** 2
**Presentation:** 3
**Contribution:** 3
**Rating:** 5
**Confidence:** 2

**Summary:**

The paper revisits the strategic facility location problem, focusing on the role of randomization in designing truthful mechanisms enhanced by machine-learned predictions. The authors build on a recent framework that refines worst-case results by incorporating potentially incorrect predictions about agents' true locations. They aim to achieve both strong consistency, which provides optimal guarantees when predictions are accurate, and near-optimal robustness, ensuring acceptable performance even with inaccurate predictions. The paper specifically investigates the power and limitations of randomization within this context, analyzing the best achievable approximation guarantees concerning the egalitarian (max) social cost measure on one- and two-dimensional Euclidean spaces.


For the single-dimensional version, the paper shows that even with the strongest possible prediction, any $1+\delta$ consistent randomized truthful mechanism can be no better than $2+\delta$-robust. This implies that the previously proposed mechanisms by Procaccia and Tennenholtz, and Agrawal et al. are both Pareto optimal. They further show that this trade-off can be achieved.

For the two-dimensional version of the problem, the authors provide a lower bound of 1.11 for all randomized mechanisms, and further show that no deterministic mechanism can combine better than 2-consistency with $1+ \sqrt{2}$​-robustness, and no randomized mechanism can combine 1-consistency with better than 2-robustness. Additionally, they present a truthful randomized mechanism that achieves 1.5 consistency and 2 robustness.

**Strengths:**

+) The paper gives pretty good results for one- and two-dimensional strategic facility locations.  In particular, the paper provides a complete characterization of Parato optimal mechanisms for the one-dimensional setting.
+) The writing is pretty good.
+) The consideration of various prediction settings may enrich the mechanism with prediction literature.

**Weaknesses:**

Some of the technical writing is unclear.  I will list some in the following sections.

- Lemma 1 should quantify $\alpha$ and $\beta$.  Theorem 3 should quantify $\epsilon$

**Questions:**

Major:

I do not follow the proof of Lemma 1. In particular, is the OnlyM mechanism at line 248 well defined? The distribution needs the knowledge of $x_L$, which is not part of the input.


Minor:
- What is formal definition of OnlyM mechanisms? Given report $\mathbf{x}$, and prediction $*$, if $x_L = \min \mathbf{x}$ and $x_R = \max \mathbf{x}$, $supp(M(\mathbf{x}, *)) = \{x_L, x_R, M\}$
- Does Lemma 1 require the original randomized mechanism to be truthful?

**Limitations:**

Yes

---

> ### Author Rebuttal · Authors · 2024-08-07
>
> Thank you for your helpful comments and suggestions.
>
> * **Reviewer:** Lemma 1 should quantify $\alpha$ and $\beta$.  Theorem 3 should quantify $\epsilon$.
>
>
> **Response:** Thank you for pointing this out. We have now quantified all these parameters and we provide the reworded statements below. Furthermore, we took this opportunity to change $\epsilon$ to $\delta$ in the statements of Theorem 3 and Theorem 4 to be consistent with our notation in Section 3.
>
> >**Lemma 1 (Reworded).** For any problem instance involving two agents with reported locations $\mathbf{x} = \langle x_L, x_R \rangle$ on the line, and any randomized mechanism that is truthful in expectation and achieves $\alpha$-consistency and $\beta$-robustness for some $1 \leq \alpha \leq \beta$ over this class of instances, there exists a randomized $OnlyM$ mechanism that is truthful in expectation and achieves the same consistency and robustness guarantees.
>
> >**Theorem 3 (Reworded).** For any $\delta > 0$, there is no deterministic truthful mechanism that is $(2-\delta)$-consistent and $(1+\sqrt{2}-\delta)$-robust, even if it is provided with full predictions $\hat{\mathbf{x}}$ (the location of each agent).
>
>
>
> >**Theorem 4 (Reworded).** For any $\delta > 0$, there is no randomized mechanism that is truthful in expectation, $1$-consistent, and $(2 - \delta)$-robust, even for two-agent instances and even if it is provided with full predictions $\hat{\mathbf{x}}$ (the location of each agent). This is tight, i.e., there exists a randomized mechanism that is truthful in expectation, $1$-consistent, and $2$-robust for two-agent instances.
>
>
>
> * **Reviewer:** "is the OnlyM mechanism at line $248$ well defined?
> The distribution needs the knowledge of $x_L$, which is not part of the input."
>
> **Response:** The line that the reviewer is referring to is actually not proposing a mechanism. The mechanism that is being analyzed throughout the proof is the one already defined in the second paragraph of the proof. The argument here starts with the distribution that the original mechanism, f, would output if the agent were to report $x'_L < x_L$. Then, it modifies this distribution to get one that yields the same cost for the agent at $x_L$ but only outputs no locations in $(x'_L, M')$ other than $x_L$ and $M$. It is this distribution that Line 248 describes and the reviewer is correct that a mechanism could not guarantee this, as it would require knowledge of $x_L$, which is private. However, this "intermediate" distribution is then further modified, starting at Line 251, to yield a probability distribution which never returns any outcomes in $(x'_L, M')$. It is this final distribution that is equivalent to the output of the mechanism defined in the second paragraph of the proof if it were provided with input $x'_L$ and $x_R$. Note that this mechanism does not need to know $x_L$, so it is a valid mechanism. Due to space limitations, we have to really compress this part of the proof, which makes it harder to read, but we will make sure to dedicate a bit more space to it in the next revision. We would also be more than happy to clarify this proof further, or even provide a fully rewritten version of the proof, if this would help convince the reviewer that there is no correctness issue here.
>
>
>
> * **Reviewer:** " Does Lemma 1 require the original randomized mechanism to be truthful?"
>
> **Response:** Yes, the original randomized mechanism is required to be truthful and we have updated the statement of the Lemma 1 to mention this explicitly (see the reworded lemma statement in our first response, above).
>
>
> * **Reviewer:** What is formal definition of $OnlyM$ mechanisms? Given report $x$, and prediction $*$, $supp (M(x, *)) = x_L, x_R, M$
>
> **Response:** No, the support of $OnlyM$ mechanisms can also include any point outside the $[x_L, x_R]$ interval. The only restriction is that their support within $(x_L, x_R)$ is just $M$ (see Line 203). To make this part clearer, we will include the following formal definition:
>
> > **Definition ($OnlyM$ mechanisms)** A mechanism $f$ is an $OnlyM$ mechanism if $P[f(\mathbf{x}) \in (x_L, x_R) \setminus \{M\}] = 0$, where $x_L$ is the leftmost reported location, $x_R$ is the rightmost report location, and $M = (x_L + x_R)/2$ is the midpoint of the two.
>
> Furthermore, to make the reduction of Lemma 1 easier to follow, we have added a figure. You can find it as Figure 1 in the PDF file that we uploaded in the general response.

---

> > ### Comment · Reviewer_vBqc · 2024-08-12
> > **Lemma 1**
> >
> > Thank you for your clarification. I understand the final ``mechanism'' in Line 251 does not output $x_L$. However, do the weights of the convex combination (probabilities) depend on $x_L$ as they need to be set to maintain the same cost for the agent at $x_L$?

---

> > > ### Author Response · Authors · 2024-08-12
> > >
> > > Thank you very much for following up. The reviewer is correct that if we wanted to ensure that the cost of the agent located at $x_L$ remains the same in that last change in the distribution, then we would need to depend on where exactly $x_L$ lies, which would be problematic. However, note that **the cost for the agent at $x_L$ is actually not maintained during this last change in the probability distribution**. As Line 253 mentions, the cost of the agent located at $x_L$ can increase (thus hurting the agent for misreporting their true location). It is only the expected coordinate of the facility location returned within $(x'_L, M')$ that is maintained, which does not require knowledge of $x_L$, and this expected coordinate is exactly what our reduction maintains.
> > >
> > > Let us provide a different way of phrasing the argument, which may hopefully make it easier to follow:
> > >
> > > Let $x'_L<x_L$ be the location reported by the agent actually located at $x_L$, and let $f$ be the original mechanism and $g$ be the OnlyM mechanism that we construct using our reduction on $f$. If we focus on the interval $(x'_L, M')$, the expected coordinate of the facility returned by the two mechanisms, $f$ and $g$, conditioned on the facility being in the interval $(x'_L, M')$, is the same; this is true by construction (our reduction maintains the expected coordinate between the reported location and the mid-point). The crucial difference between the two mechanisms is that although $f$ may return any facility location in the $(x'_L, M')$ interval, mechanism $g$ returns the same expected coordinate using only $x'_L$ and $M'$ in its support. Our proof shows that the cost of an agent located at **any point $x_L$** in the $(x'_L, M')$ interval is weakly higher in $g$ than it is in $f$, thus maintaining the truthfulness guarantee. Intuitively, this is true because the two mechanisms return the same expected coordinate in that interval, but $g$ only returns the extreme points of the interval, which hurt the agent at $x_L$ the most. More formally, the proof in our submission proves this intuition by first constructing an "intermediate" distribution (Line 251) which has *the same cost as $f$ for the agent located at $x_L$ and also has the same expected coordinate as both $f$ and $g$*. Then, we observe that $g$ has weakly higher cost for the agent located at $x_L$ compared to this "intermediate" distribution, because the latter may assign some probability on $x_L$, whereas $g$ instead distributes that probability between $x'_L$ and $M'$, which increases the cost of the agent located at $x_L$.
> > >
> > > Hopefully this addresses the reviewer's concern. If not, we would be happy to provide additional details. We acknowledge that the subtleties discussed above (which, we believe, make the result technically demanding and interesting) were written too densely, due to space limitations. We would be happy to either unpack this proof further within the main body of the paper, or to include a more detailed proof in the full version or the appendix.

---

> ### Comment · Reviewer_vBqc · 2024-08-12
>
> I see. The intermediate construction confuses me. Alternatively, you may use the convexity argument of the distance function.  Thank you for your clarification.  I do not have further questions.

---

> > ### Author Response · Authors · 2024-08-13
> >
> > That sounds reasonable. We would be happy to instead present this transition from distribution $f$ to distribution $g$ by gradually replacing the probability that $f$ may assign to points in $(x'_L, M')$ with probability assigned to just $x'_L$ and $M'$, while maintaining the same expected location. During this process, the convexity of the cost function of any agent located within the interval can be used to verify truthfulness. This way, we can skip the intermediate distribution that the reviewer finds confusing.

---

### Official Review · Reviewer_zBmc · 2024-07-14

**Soundness:** 4
**Presentation:** 4
**Contribution:** 3
**Rating:** 7
**Confidence:** 2

**Summary:**

The authors study the problem of designing incentive-compatible mechanisms for the facility location game, in particular a learning-augmented variant of ABGOT22 where an oracle returns (potentially inaccurate) predictions about optimal solutions. The paper focuses on considering when predictions are available for the preferences of individual participants/which participants are extremal.

In the 1d setting, the authors show that randomizing between known algorithms is already optimal---allowing predictions about individual preferences cannot result in a mechanism that dominates prior algorithms which depend at most only on optimal facility predictions. In the 2d setting, the authors show that a slight improvement over state-of-art is possible by only assuming prediction access to identification of extremal agents; the authors also provide a few factoids e.g. minor improvement in approximation ratio lower bound for the general (non learning-augmented) setting.

**Strengths:**

The paper's results are novel, cleanly presented, and indeed strictly improve upon several bounds in literature. The results on the line are not particularly surprising---e.g. Theorem 1 is just a short proof-by-contradiction given the reduction of Lemma 1 which seems to be largely implied by PT13---but are well-written and an accessible warmup to the plane results, which appear to be the main contributions of the paper.
In terms of the results on the plane, although Theorem 6 also is not particularly surprising---circles have three degrees of freedom and one should expect mechanisms to really only depend on the reports of extremal agents, which on a plane should always be about three in practice---verifying the result is still non-trivial and the paper gives a clean exposition and proof.
The paper also benefits from a collection of various observations; for example, the lower bound of Theorem 2 seems to just be a short note that PT13's 1d lower bound can be easily lifted to 2d.
Overall the paper is also very well written and gentle.

**Weaknesses:**

I only have minor notes on polishing the writing. Theorem 5 can make more explicit that an assumption is being made on the number of extreme agents---it was a bit confusing when initially skimming the paper. It would also be helpful to clarify if Lemma 1 is novel; from my reading it's just an consequence of the linearity of in-expectation honesty and seems to be implied by PT13---if this is correct, it would be helpful to note in the text. Also, L372-L377 were a bit difficult to understand without drawing it out graphically by hand; perhaps a diagram would help readability.

**Questions:**

See weaknesses section.

**Limitations:**

Yes the authors have addressed any negative societal impacts.

---

> ### Author Rebuttal · Authors · 2024-08-07
>
> Thank you for your helpful comments and suggestions. We actually think that our 1-dimensional results are quite significant on their own and much more than a warm-up for the two-dimensional ones. We provide some justification for this fact in response to the reviewer's question below.
>
>
>
> * **Reviewer:** "It would also be helpful to clarify if Lemma 1 is novel; from my reading it's just an consequence of the linearity of in-expectation honesty and seems to be implied by PT13---if this is correct, it would be helpful to note in the text."
>
> **Response:** Lemma 1 is, indeed, novel, and we do not see how it could be implied by any argument in PT13. To maintain the robustness, consistency, and truthfulness guarantees throughout this reduction, we do make use of linearity of expectation, however, there is much more to this proof than that. Specifically, note that there can be an infinite number of ways to rearrange the probability that a mechanism assigns to outcomes in $(x_L, x_R)$ to get an OnlyM mechanism without affecting the expected cost of the two agents located at $x_L$ and $x_R$. For example, we could first replace the outcomes in $(x_L, x_R)$ with a single outcome whose coordinate corresponds to the expected coordinate of these outcomes, and then express this single outcome as a convex combination of $M$, $x_L$, and $x_R$ (and there can many such convex combinations). However, maintaining the robustness and consistency guarantees limits the ways in which we can do that, and the particular way in which we rearrange the probability is really crucial for us to maintain the very delicate truthfulness property. The part of our argument that really leverages this subtlety starts at Line 244, and it argues that if the agent located at $x_L$ were to lie and report a location further away from the other agent (i.e., $x'_L < x_L$), then the amount of probability that the modified mechanism would assign to $x'_L$ instead of $x_L$ would penalize the agent for the lie more than any benefit that the agent could get from this lie.
>
> Although this is one of the more technical parts of the proof, we had to really compress it in order to fit the results within the page limit, but we would be happy to emphasize this part of the proof much more in the next revision of the paper and to explain how some alternative approaches would fail to maintain truthfulness, robustness, or consistency.
>
> Furthermore, to make the reduction of Lemma 1 easier to follow, we have added a figure. You can find it as Figure 1 in the PDF file that we uploaded in the general response.
>
>
>
>
> * **Reviewer:**"Theorem 5 can make more explicit that an assumption is being made on the number of extreme agents---it was a bit confusing when initially skimming the paper."
>
> **Response:** We agree with the reviewer. We will rephrase the statement to more clearly emphasize this assumption, as follows:
>
> >**Theorem 5 (Reworded).** Consider any instance that has only two extreme agents, i.e., $k = 2.$ Then, given predictions $\hat{\mathbf{e}} = \langle e_1, e_2 \rangle$ regarding the IDs of the two extreme agents, there exists a randomized mechanism that is truthful in expectation and achieves $1.5$ consistency and $2$ robustness for any number of agents.
>
>
>
>
> * **Reviewer:**"Also, L372-L377 were a bit difficult to understand without drawing it out graphically by hand; perhaps a diagram would help readability."
>
> **Response:** We agree with the reviewer. We added a figure to make the proof of Theorem $6$ easier to follow. You can find it as Figure 3 in the PDF file that we uploaded in the general response.

---

> > ### Comment · Reviewer_zBmc · 2024-08-09
> > **Response**
> >
> > I appreciate the clarifications and look forward to the revisions. I also appreciate the clarifications regarding Lemma 1---I didn't actually feel that the proof was too compressed but agree that it could be clarified (the diagram is nice). I maintain my positive review otherwise.

---

> > > ### Author Response · Authors · 2024-08-12
> > >
> > > Thank you very much. Note that in our latest response to reviewer vBqc, we now unpack that last part of the proof of Lemma 1 a bit further, which points to some of the subtleties that make this proof non-trivial and one of our main contributions. We thought we should point this out to you as well, given your original hesitation regarding the significance of this result.

---

### Author Rebuttal · Authors · 2024-08-07

We thank all the reviewers for the time and effort they dedicated to evaluating our paper. We have provided individual responses for their questions and would be more than happy to provide additional details if any of these responses do not fully address the reviewers' concerns. Attached to this response you can find a PDF file that provides new figures that we have added to enhance the readability of some of our proofs, following the reviewer's suggestions and feedback.

---

### Decision · Program_Chairs · 2024-09-25

**Decision:**

Accept (poster)

**Comment:**

The paper studies strategic facility location in 1 and 2 dimensions with expert advice. Specifically, it considers achieving a consistent and robust (randomized) mechanism, focusing on the Egalitarian cost.

The main **contributions** are:
1. They are characterizing the Pareto frontier for the line. The authors show that two previous papers provide Pareto optimal solutions. Flipping coins between them can characterize a Pareto frontier (Proposition 1 + Theorem 1).
2.	They provide negative (impossibilities) and positive (Mechanism 1) results for the plane.

The main **strengths** I see in this paper are:
1. Strong technical results on 1d, answering questions from previous papers, and initial results for 2d.
2.	Relating to a vibrant line of work on mechanisms with expert advice.

**The first weakness** (and the less important one) is the narrow impact of publishing this paper on the community. The paper belongs to the intersection of the “algorithms with expert advice” and the “strategic facility location,” which is relatively small. I consider this as a weakness since the techniques the authors use (seem to) apply to this intersection only.

The second and **main weakness** I see in this paper, which also surfaced in the reviews, is the quality of the technical writing. The paper includes typos and cumbersome writing in mathematical statements, which are usually a red flag. Beyond that, some proofs are hard-to-impossible to follow (e.g., Lemma 1).

The first weakness cannot be overcome. As for the second weakness, the authors have done an excellent rebuttal addressing reviewers’ comments about this concern. They have rephrased statements, facilitated proofs, and attached figures to ease readability.

I want to note that in private communication with the other reviewers, they expressed their lack of confidence about the extent of this paper’s contribution. Furthermore, no reviewer wanted to champion it.

After carefully reading the paper, the reviews, and the rebuttal, I see this paper as borderline, yet **I tend to accept it**. However, I understand it is a borderline paper and might be rejected due to the low acceptance rate.